# A comparison of methods to estimate vertical land motion trends from GNSS and altimetry at tide gauge stations

Marcel Kleinherenbrink[1], Riccardo Riva[1], and Thomas Frederikse[1]

[1]Department of Geoscience and Remote Sensing, Delft University of Technology, P.O. Box 5048, 2600 GA Delft, The Netherlands

*Correspondence to:* Marcel Kleinherenbrink (m.kleinherenbrink@tudelft.nl)

**Abstract.** Tide-gauge (TG) records are affected by Vertical Land Motion (VLM), causing them to observe relative instead of geocentric sea level. VLM can be estimated from Global Navigation Satellite System (GNSS) time series, but only a few TGs are equipped with a GNSS receiver. Hence, (multiple) neighouring GNSS stations can be used to estimate VLM at the TG. This study compares eight approaches to estimate VLM trends at 570 TG stations using GNSS, by taking into account all GNSS trends with an uncertainty smaller than 1 mm yr$^{-1}$ within 50 km. The range between the methods is comparable with the formal uncertainties of the GNSS trends. Taking the median of the surrounding GNSS trends shows the best agreement with differenced altimetry - tide gauge (ALT-TG) trends. An attempt is also made to improve VLM trends from ALT-TG time series. Only using highly correlated along-track altimetry and TG time series, reduces the standard deviation of ALT-TG time series up to 10%. As a result, there are spatially coherent changes in the trends, but the reduction in the RMS of differences between ALT-TG and GNSS trends is insignificant. However, setting correlation thresholds also acts like a filter to remove problematic TG time series. This results in sets of ALT-TG VLM trends at 344-663 TG locations, depending on the correlation threshold. Compared to other studies, we decrease the RMS of differences between GNSS and ALT-TG trends (from 1.47 to 1.22 mm yr$^{-1}$), while we increase the number of locations (from 109 to 155), Depending on the methods the mean of differences between ALT-TG and GNSS trends varies between 0.1-0.2 mm yr$^{-1}$. We reduce the mean of differences by taking into account the effect of elastic deformation due to present-day mass redistribution. At varying ALT-TG correlation thresholds, we provide new sets of trends for 759 to 939 different TG stations. If both GNSS and ALT-TG trend estimates are available, we recommend to use the GNSS trend estimates, because residual ocean signals might correlate over long distances. However, if large discrepancies (> 3 mm yr$^{-1}$) between both methods are present, local VLM differences between the TG and the GNSS station are likely the culprit and therefore is is better to take the ALT-TG trend estimate. Especially GNSS estimates where only a single GNSS station and no ALT-TG estimate is available, might still require some inspection before they are used in sea level studies.

## 1 Introduction

Tide Gauges (TGs) measure local relative sea level, which means that they are affected by geocentric sea level, but also by Vertical Land Motion (VLM). Knowing VLM at TGs is essential to convert the observed sea level into a geocentric reference

frame, in which among others satellite altimeters operate. TGs used in sea level reconstructions also require a correction for VLM. The mean of VLM at TGs is not equal to that of the basin, and therefore local VLM estimates are required to get an accurate estimate of ocean volume change. The models for large scale VLM processes, such as Glacial Isostatic Adjustment (GIA) and the elastic response of the Earth due to present-day mass redistribution, are becoming more accurate. TGs are often

only corrected for the GIA signal, which typically reaches values of 10 mm yr$^{-1}$ in Canada and Scandinavia (*Gutenberg et al.*, 1941). The elastic deformation due to present-day mass redistribution is often ignored. However, elastic deformation is becoming larger due to the increasing rate of Greenland's ice mass loss, and to a lesser extent other processes. Trends at TGs are also affected by a large number of other local signals, including water storage, postseismic deformation and anthropogenic activities (*Hamlington et al.*, 2016; *Wöppelmann and Marcos*, 2016). Since the local VLM processes cannot be captured by

models, and the large-scale processes contain large uncertainties, observations of VLM at TGs are essential.

One method to estimate VLM at TGs uses geodetic Global Positioning System (GPS) receivers at fixed stations or Doppler Orbitography and Radiopositioning Integrated by Satellite (DORIS) observations. Since many other navigation satellites are currently providing range estimates as well, we will refer to the GPS stations as Global Navigation Satellite System (GNSS) stations. Most studies compute GNSS VLM at TG stations from one of the datasets by University of La Rochelle (ULR)

(*Wöppelmann et al.*, 2007; *Pfeffer and Allemand*, 2016; *Wöppelmann et al.*, 2014; *Wöppelmann and Marcos*, 2016). Even though ULR contains several GNSS solution inland, its main focus is the coastal zone. Currently, 754 GNSS stations are processed in the ULR6 database. A more extensive database with approximately 14000 GNSS is processed by the Nevada Geodetic Laboratory (NGL). They use a different processing procedure to estimate trends from time series, which makes trends less vulnerable to jumps (*Blewitt et al.*, 2016). A statistical comparison between several GNSS solutions was recently

made by *Santamaría-Gómez et al.* (2017). They concluded that the number of stations in the NGL database was larger, but that the differences between neighbouring stations was significantly larger than the Jet Propulsion Laboratory (JPL) and ULR6 trend estimates. They also discussed systematic errors due to differences in the origin of the reference frames, which were in the order of 0.2 mm yr$^{-1}$ globally. Furthermore, they found that the local VLM uncertainty at tide gauge was increased by $4 \times 10^{-3}$ mm yr$^{-1}$ per kilometer distance between the TG and the GNSS station (*Santamaría-Gómez et al.*, 2017). Most studies use the trends

of either co-located GNSS stations or the closest GNSS station or the mean of all GNSS stations within a radius of several tens of kilometers (*Santamaría-Gómez et al.*, 2014; *Pfeffer and Allemand*, 2016). Only *Hamlington et al.* (2016) involved a more complex GNSS post-processing procedure using NGL trends, based on a combination of spatial filtering, Delaunay triangulation and median weighting. One way to quantify the accuracy of GNSS-based VLM trends at TGs is to compute the spread of individual geocentric sea level estimates or the spread of geocentric sea level between regions (*Wöppelmann and*

*Marcos*, 2016). The spread of regional trends reduced from 0.9 mm yr$^{-1}$ in the ULR1 solution (*Wöppelmann et al.*, 2007) to 0.5 mm yr$^{-1}$ in the ULR5 solution (*Santamaría-Gómez et al.*, 2012; *Wöppelmann et al.*, 2014), which is approximately the expected residual climatic signal. Any further improvements in the GNSS trends require therefore another validation technique.

A second way to observe VLM at TGs, to overcome the limitations of sparsely distributed GNSS network, is differencing satellite altimetry and TG time series, which we will refer to as ALT-TG time series from here on. Initially, the ALT-TG

time series were used to monitor the stability of satellite altimeters for the Global Mean Sea Level (GMSL) record, which is

currently gauranteed up to 0.4 mm yr$^{-1}$ (*Mitchum*, 1998, 2000). The first study to infer VLM trends from ALT-TG time series was *Cazenave et al.* (1999). Based on the method of *Mitchum* (1998) they compared ALT-TG to DORIS at six stations. Later, several studies were conducted on regional and global scale of which an overview is given by *Ostanciaux et al.* (2012). The first study to estimate more than 100 VLM trends (*Nerem and Mitchum*, 2002) obtained error bars for 60 of 114 TGs smaller than 2 mm yr$^{-1}$. However, they noted that the TGs should be inspected on a case-by-case basis to determine if the result was truly VLM. *Ostanciaux et al.* (2012) increased the number of ALT-TG VLM trend estimates sixfold to 641, but it included some outliers with trends above 20 mm yr$^{-1}$. They also made a comparison between their study and several earlier studies. The best agreement was found over a small set of 28 tide gauges, where the results of *Ostanciaux et al.* (2012) differed from (*Ray et al.*, 2010) by an RMS of 1.2 mm yr$^{-1}$.

Recently, several studies have compared the GNSS trends to those of ALT-TG globally (*Santamaría-Gómez et al.*, 2014; *Wöppelmann and Marcos*, 2016; *Pfeffer and Allemand*, 2016). Several other studies did an equivalent comparison with DORIS and ALT-TG for a limited number of stations (*Cazenave et al.*, 1999; *Nerem and Mitchum*, 2002; *Ray et al.*, 2010). While the older studies primarily used along-track data from the Jason (TOPEX/POSEIDON (TP), Jason-1 (J1) and Jason-2 (J2)) series of satellite altimeters, the latest studies used preprocessed grids and *Wöppelmann and Marcos* (2016) made a comparison between several gridded products and one along-track dataset. All recent studies used ULR5 GNSS trends for comparison. The best results were obtained with an interpolated altimetry grid provided by AVISO (*Pujol et al.*, 2016), yielding a median of differences of 0.25 mm yr$^{-1}$ with an RMS of 1.47 mm yr$^{-1}$ based on a comparison at 107 locations (*Wöppelmann and Marcos*, 2016). It is important to note that the time series for all sites were visually inspected, primarily to remove those with non-linear behaviour. Additonally, the corresponding correlation between altimetry and TG time series were found to be highest for AVISO. *Pfeffer and Allemand* (2016) did not apply visual inspection and obtained a comparable result for 113 stations (an RMS of 1.7 mm yr$^{-1}$), while only incorporating GNSS trends from stations within 10 km from the tide gauge.

This study aims to further reduce the discrepancies between GNSS and ALT-TG trends, while increasing the number of trend pairs. To do this, we will apply several steps to improve the VLM estimates at tide gauges. First of all, the number of reliable trend estimates are increased by using the GNSS trends from the larger NGL database. Most TGs will neighbour multiple GNSS stations for which several methods are applied to determine the best procedure. Correlations between altimetry and TG time series are exploited to reduce residual ocean variability, which is often present in ALT-TG time series (*Vinogradov and Ponte*, 2011). The reduction in ocean variability should lead to more reliable ALT-TG VLM trends. Correlation thresholds additionally function as a filter, to remove time series that are uncorrelated due to differences in ocean signals, possible (undocumented) jumps in the TG time series, or interannual VLM signals that cannot be separated from the ocean signal (*Santamaría-Gómez et al.*, 2014). Additionally, we address the problem of contemporary mass redistribution on trends over different time spans using a fingerprinting method.

## 2 Data and Methods

In this section, we describe the processing procedures for deriving GNSS and ALT-TG VLM trends for comparison at TG locations. First, we will address the estimation of GNSS trends at the TG locations. The estimation of ALT-TG differenced trends is discussed in several steps. We briefly discuss the selection of the tide gauges. After that we will discuss the altimetry processing procedures. We briefly review the Hector software (*Bos et al.*, 2013) for the estimation of trends from differenced ALT-TG time series. Eventually, trend corrections for contemporary mass redistribution using fingerprinting methods are described.

### 2.1 GNSS trends

The trend estimation at tide gauges primarily deals with two problems. First, a trend is estimated from a GNSS time series, which contains an autocorrelated noise signal, and often undocumented jumps. We use pre-computed trends, of which the procedure is briefly reviewed in Sect. 2.1.1. Second, many GNSS stations are not directly co-located to the TG station. Regular leveling campaigns, to monitor the relative VLM between the TG and the GNSS stations, after often absent. Therefore, the assumption is made that both locations are affected by the same VLM signal. When multiple GNSS receivers are present in the vicinity of the tide gauge, a method is required to estimate a single VLM trend from multiple GNSS stations. This is discussed in Sect. 2.1.2.

### 2.1.1 GNSS trend estimation

To obtain VLM trends at TGs, often the products of the Université de La Rochelle (ULR) are used. ULR versions 5 and 6 make use of the Create and Analyze Time Series (CATS) software (*Williams*, 2008), which is able to estimate trends and errors from time series, taking into account temporally correlated noise. It has the advantage that it computes a more realistic trend uncertainty. The software is also able to estimate and detect discontinuities that occur due to earthquakes and equipment changes. Even though a large proportion of the trend estimates have formal accuracies better than 1 mm/yr, undetected discontinuities might bias the estimated trends (*Gazeaux et al.*, 2013).

In this study the results of NGL (*Blewitt et al.*, 2016) are used. *Blewitt et al.* (2016) proposed the Median Interannual Difference Adjusted for Skewness (MIDAS) approach, which is based on the Theil-Sen estimator. The procedure estimates trends from couples of daily data points separated by 365 days. It then removes all estimates outside two standard deviations, which are computed by scaling the Median of Absolute Devations (MAD) by 1.4826 (*Wilcox*, 2005), with respect to the median of the trend couples. Afterwards, a new median is computed, which serves as the trend estimate. *Blewitt et al.* (2016) demonstrated that MIDAS has a smaller equivalent step detection size than methods which included step detection, such as those computed by CATS and used by ULR5. Besides the advantage of detecting smaller jumps, approximately 14000 GNSS time series are processed, which is almost 20 times more than ULR6. Unlike *Wöppelmann and Marcos* (2016), no manual screening is applied to the time series or trends.

### 2.1.2  Trend estimation at tide gauges

Despite several recommendations to co-locate GNSS receivers with TGs, currently only a few have a record that ensures a trend uncertainty of 1 mm yr$^{-1}$ or better. Therefore we take all stations into account that are within 50 km from a TG, provided that the standard deviation on the trend is lower than 1 mm yr$^{-1}$ as estimated from the MIDAS algorithm. The threshold on
the standard deviation ensures that most records containing large non-linear effects, due to for example earthquakes and water storage changes, are removed from the analysis. Other studies used ranges from 10 km (*Pfeffer and Allemand*, 2016) up to 100 km (*Hamlington et al.*, 2016). At 100 km the error due relative VLM trends increases substantially, on average with more than 0.5 mm yr$^{-1}$ (*Santamaría-Gómez et al.*, 2017) for the NGL estimates, while taking a range of 10 km reduces the number of trends substantially. Therefore the range is set to 50 km, but comparable results are found for 30 and 70 km yielding a different
number of trends (not shown).

Most studies simply average all neighbouring TG trends or take the trend from the closest station. However, many other and possibly better, techniques are possible. We compare trends from several approaches in Sect. 3.1 and with the ALT-TG trends in Sect. 3.3. In total eight different approaches are considered. The first two involve all of the trends at neighbouring GNSS stations by computing their mean [1] and median [2]. Method [1] is among others applied by (*Frederikse et al.*, 2016)
for regional sea level reconstructions. One of the most frequently applied approach uses the trend at the closest station [3]. It is used in two recent studies by *Santamaría-Gómez et al.* (2012) and *Pfeffer and Allemand* (2016). We also investigate inverse distance weighting [4] in which the trend $\frac{dh_{TG}}{dt}$ is estimated as:

$$\frac{dh_{TG}}{dt} = \frac{\sum \frac{1}{d_i} \frac{dh_i}{dt}}{\sum \frac{1}{d_i}}, \tag{1}$$

where $d_i$ and $\frac{dh_i}{dt}$ respresent the distance to the tide gauge station and the trend at GNSS station $i$. We also use the GNSS trends
based on the longest time series [5] and smallest error [6] from stations within the 50 km radius. The seventh approach involves weighting with the variances $\sigma_i^2$ of the trends [7], such that:

$$\frac{dh_{TG}}{dt} = \frac{\sum \frac{1}{\sigma_i^2} \frac{dh_i}{dt}}{\sum \frac{1}{\sigma_i^2}}. \tag{2}$$

And the last method [8] takes into account spatial dependency and trend uncertainty by combining methods [4] and [7], i.e. by weighting with the variance and with the distance, so that:

$$\frac{dh_{TG}}{dt} = \frac{\sum \frac{1}{\sigma_i^2 d_i} \frac{dh_i}{dt}}{\sum \frac{1}{\sigma_i^2 d_i}} \tag{3}$$

Method [8] is a variant to the technique used in the altimeter calibration study of *Watson et al.* (2015). Note that the uncertainties range mostly between 0.7-1 mm yr$^{-1}$ and therefore method [8] is more sensitive to the distance from the TG than to the variance of the GNSS trends. The distance weights used in methods [4] and [8] quickly decrease with distance, effectively reducing the number of GNSS trends involved in the estimate. In several studies the method to estimate VLM trends at tide gauges from
GNSS is not documented.

## 2.2 Tide gauge time series

Monthly TG data are obtained from the PSMSL database (*Holgate et al.*, 2013). All time series flagged after 1993 are removed. Any observations that are outside of 1 meter from the mean are considered outliers and removed from the data. This number is similar to our altimetry sea level threshold and based on the criterion used by NOAA for their global mean sea level estimates (*Masters et al.*, 2012). To be consistent with the altimetry observations, we apply a Dynamic Atmosphere Correction (DAC) consisting of a low-frequency inverse barometer correction and short-term wind and pressure effects *Carrère and Lyard* (2003). Initially, we consider all TGs with at least 10 years of valid data.

## 2.3 Differenced ALT-TG time series

*Wöppelmann and Marcos* (2016) obtained the smallest standard deviation in the differenced time series by averaging grid cells within 1 degree from the TG using the AVISO interpolated product. The results obtained by taking the most correlated grid point from AVISO within 4 degrees around the TG increased the standard deviation. *Wöppelmann and Marcos* (2016) obtained lower correlations by averaging Goddard Space Flight Center (GSFC) along-track altimetry measurements within a radius of 1 degree from the TG. Note that the AVISO grid is constructed using correlation radii of 50-300 km (*Ducet et al.*, 2000) and it includes measurements from all altimetry satellites, not only the Jason series. The AVISO grid therefore effectively averages over a much larger radius around the TG and it includes data from more satellites. The larger uncorrelated noise using GSFC compared to AVISO, as shown by the combination of the increased RMS and the spectral index (*Wöppelmann and Marcos*, 2016), is therefore likely an effect of the limited number of GSFC altimetry measurements. However, using the large effective radius of AVISO, data far away from the TG are included, which might not correlate with the sea level signal at the TG. This can result in a remaining ocean signal in ALT-TG time series, which contaminates the VLM trend estimates.

**Table 1.** List of geophysical corrections and orbits applied in this study.

| Satellite | T/P | Jason-1&2 |
|---|---|---|
| Orbits | CCI | GDR-E |
| Ionosphere | Smoothed dual-frequency | |
| Wet troposphere | Radiometer | |
| Dry troposphere | ECMWF | |
| Ocean tide | GOT4.10 | |
| Loading tide | GOT4.10 | |
| Solid Earth tide | Cartwright | |
| Sea state bias | CLS | |
| Mean sea surface | DTU15 | |
| Dynamic atmosphere | MOG2D | |

To overcome the limitations of gridded products, we work with along-track data and exploit the correlations between sea level at the satellite measurement location and at the TG on interannual and decadal scales by using a low-pass filter. We start by creating sea level time series every 6.2 km along-track using the measurements from TP, J1 and J2 from the RADS database (*Scharroo et al.*, 2012) between 1993-2015. In order to get a consistent set of altimetry observations, the same geophysical correction are used for all satellites, as are given in Table 1. All time series within 250 km from the TG are taken into account. This radius is larger than the open ocean correlation distances used by *Ducet et al.* (2000) and *Roemmich and Gilson* (2009), except for the equatorial region where the correlation scales become much larger. At distances larger than 250 km, one will still find some highly correlated signals, but the trends caused by large scale processes like GIA and present-day mass redistribution will differ from that at the TGs. It also ensures that at least one ground track of the altimeters is within the range of the tide gauge at the equator. Reducing the 250 km radius leads to a decreased number of trends.

Additionally, intermission biases between TP-J1 and J1-J2 are removed. *Ablain et al.* (2015) revealed a large dependence of the intermission biases on the latitude. For the J1-J2 differences, a single polynomial is estimated through the differences between the sea level observations of both instrument, such that the correction $\Delta h_{sla,ib}(\lambda)$ becomes:

$$\Delta h_{sla,ib}(\lambda) = c_0 + c_1 \cdot \lambda + c_2 \cdot \lambda^2 + c_3 \cdot \lambda^3 + c_4 \cdot \lambda^4, \tag{4}$$

with $\lambda$ the latitude of the altimetry observations. For the TP-J1 differences, separate polynomials are estimated for four latitude regions and the ascending/descending tracks (*Ablain et al.*, 2015). The values for the parameters $c_n$ are given in Table 6. More details on the computation procedure are found in Appendix A.

The Jason satellite series samples sea level every ten days, hence we average monthly 3-4 measurements in order to make a first set of time series that is compatible with the monthly TG observations. As for the case of the TG monthly solutions, observations more than 1 m from the mean sea surface are removed and the time series should have at least 10 years of valid observations. Additionally, a second set of time series at each satellite measurement location is created, by applying a yearly moving-average filter. This second set of altimetry time series is correlated with a yearly low-pass filtered version of the TG series, in order to test whether their signals match on interannual and longer time scales. The yearly moving-average filter allows to suppress the noise present in individual altimetry measurements. The full pole tide from RADS (which contains a solid Earth, loading and ocean tide as in *Desai et al.* (2015)) is subtracted from both time series before correlation, whereas for the TG time series we restore the solid Earth pole tide as computed in *Desai et al.* (2015). The loading tide is at its maximum only a few millimeters, which has no significant effect on the interannual correlation, and is therefore not restored. We also remove residual annual and semi-annual cycles and a linear trend before correlation, because the yearly moving-average filter has side-lobes causing these seasonal signals to be partly retained. Other longer filters are considered to reduce the side lobes, but they would introduce larger transient zones. An iterative procedure removes sea surface heights outside of 3 RMS up to a maximum of 10 % of the observations. The outlier removal is primarily implemented to remove any spurious data present in the RADS database. It is unlikely that more than 10 % of the observations contain processing problems or outliers due to extreme events. If more observations would be discarded, high correlations might not represent corresponding ocean signal

anymore. The result is a set of correlations that indicate which altimetry sea level time series resemble the TG time series on interannual time scales and longer.

The monthly low-pass filtered altimetry time series are kept, if the corresponding correlation from yearly low-pass filtered time series are above a certain threshold. We combine the remaining monthly altimetry time series, to get one averaged altimetry time series per TG. Alternatively, we also use the correlations as weights, to get one correlation-weighted altimetry time series per tide gauge. In this case the monthly low-pass filtered time series are weighted by their corresponding correlation, then added together and accordingly normalized, so that the weights sum up to one. The resulting time series are subtracted from the TG time series if there are at least ten altimetry time series with a correlation above the threshold. The resulting differenced ALT-TG time series with less than 15 years of valid observations are further discarded. This last requirement is due to the fact that remaining ocean signals can still affect the estimated trends significantly. An example of the reduction of variability due to correlation thresholds and weighting is shown in Fig. 1. The white noise in the unfiltered time series is reduced in the red curve, however the opposite might happen if the number of altimetry time series decreases. Most important is to note that there is a strong reduction in the variance of temporally correlated residuals, represented here by the low-pass filtered time series. Correlated residual signal can strongly affect the estimated trend, especially in areas with large variability due to interannual event like ENSO. Note that for the differentiation of the time series only the solid Earth part of the pole tide is added to the TGs, but this time as is done in the IERS2010 conventions (*Petit and Luzum*, 2010), such that the trends are consistent with those of the GNSS data. The main difference is that the altimetry pole tide correction of *Desai et al.* (2015) is computed with respect to linearly drifting mean pole, while in the IERS conventions the mean pole location is modelled as a third order polynomial. If the pole tide is not taken into account consistently, it can introduce biases of 0.1 mm yr$^{-1}$ (*Santamaría-Gómez et al.*, 2017). Since the change rate of the mean pole is non-linear, this will introduce trend biases if the time spans between GNSS and altimetry do not match. The drift of the mean pole is caused by redistribution of mass in the Earth system. This is corrected for using the mass-redistribution fingerprints discussed in Sect. 2.5, which are computed using a model that includes elastic responses and rotation changes. The drifting mean pole is primarily captured by the $C_{21}$ and $S_{21}$ spherical harmonic coefficients (*Wahr et al.*, 2015).

## 2.4 Differenced ALT-TG trends

The ALT-TG time series have a monthly resolution, so they contain less observations, and they exhibit substantial interannual variability. These time series are therefore less suitable to be processed with the MIDAS algorithm used to compute GNSS trends. For the computation of the ALT-TG trends and the corresponding standard deviation, we fit a power-law in combination with a white noise model by using the Hector software (*Bos et al.*, 2013). The spectrum of the white noise is flat, while the spectrum of power-law noise, $P(f)$, decays with frequency and is given by (*Bos et al.*, 2013):

$$P(f) = \frac{1}{f_s^2} \frac{\sigma^2}{(2\sin(\pi f/f_s))^{2d}}, \tag{5}$$

where $f_s$ is the sampling frequency, $\sigma$ the power-law noise scaling factor and $d$ links to the spectral index $\kappa$ in *Wöppelmann and Marcos* (2016) by $\kappa = -2d$. The value of $d$ affects the effective number of autoregressive parameters (*Bos et al.*, 2013).

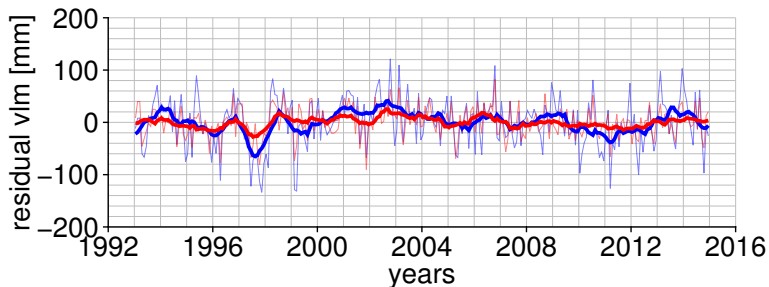

**Figure 1.** Time series of ALT-TG differenced VLM at Winter Harbour. After averaging or weighting with the correlation a moving-average filter is applied to visualize the remaining interannual variability. In blue: without a threshold on the correlation and without correlation weighting. In red: with a threshold of 0.7 for the correlation and with correlation weighting. In the background the time series without the moving-average filter applied.

This is required to capture the temporal correlation in the ALT-TG time series as shown by Fig. 2 in which the low-pass filtered time series give an idea of the memory in the system. In order to handle several weakly non-stationary ALT-TG time series we use the function 'PowerlawApprox', which uses a Toeplitz approximation for power-law noise (*Bos et al.*, 2013).

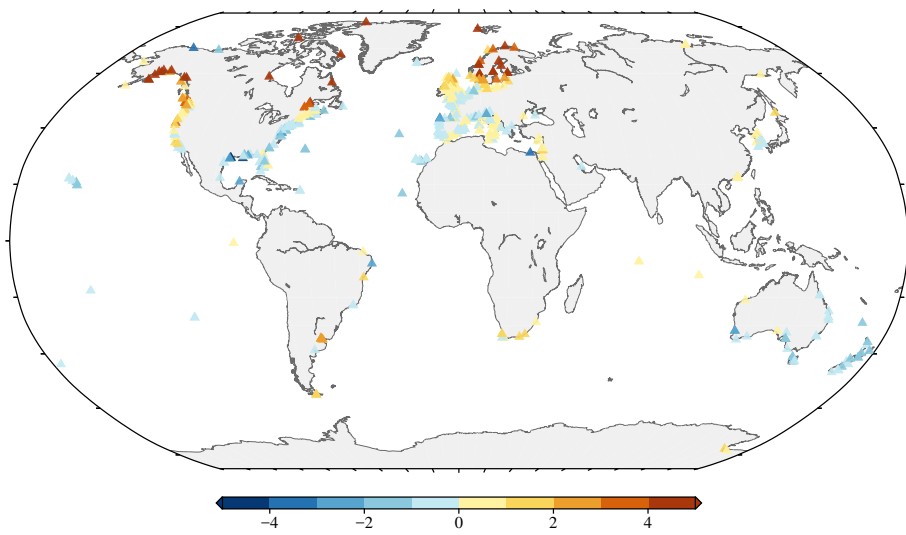

**Figure 2.** VLM (mm/yr) at TGs using the median of the neighbouring trends.

## 2.5    Contemporary mass redistribution

5    The trends estimated from GNSS time series are computed over different time spans than the ALT-TG trends and will be affected by non-linear VLM induced by elastic deformation due to present-day ice melt and changes in land hydrology storage (*Riva et al.*, 2017). To quantify those non-linear VLM signals, the response to mass redistribution is computed using a

fingerprinting method at yearly resolution. We take into account the loads of Greenland, Antarctica and glacier mass loss, the effects of dam retention and hydrological loads. A detailed description of the input loads is given in (*Frederikse et al.*, 2016). To estimate the fingerprints of VLM, the sea level equation is solved, including the rotational feedback (*Farrell and Clark*, 1976; *Milne and Mitrovica*, 1998). Since not all load information for 2015 and 2016 is available yet, we will limit the time series of ALT-TG up to 2015. Some GNSS trends are estimated from time series that span beyond 2015. Therefore we linearly extrapolate the fingerprint data, if necessary, to 2015 and 2016 based on the difference between years 2013 and 2014.

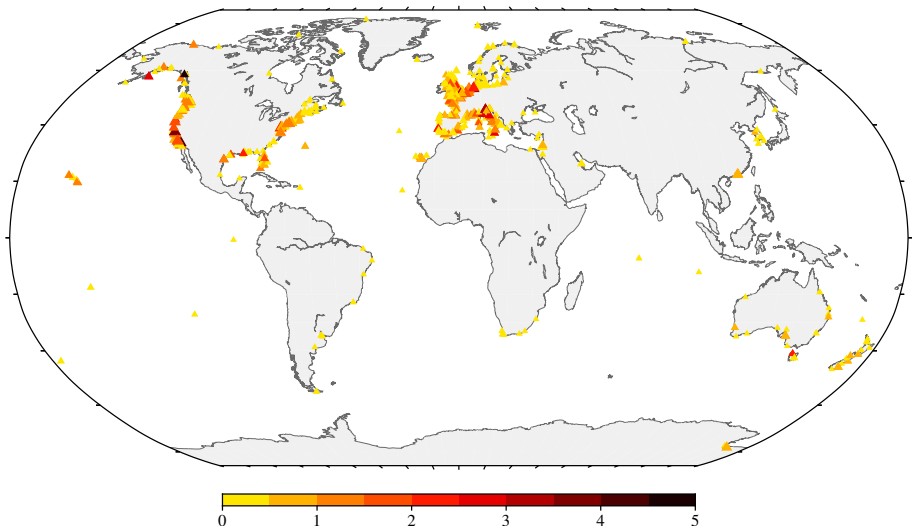

**Figure 3.** Range (mm/yr) of VLM estimates at TGs using eight different approaches. The size of the symbols indicates the number of GNSS trends available (with a maximum of ten).

## 3  Results

This section first addresses the trends obtained from GNSS stations. The averaging methods are discussed and the NGL trends are compared to those of ULR5. Then the results of the correlation-weighted ALT-TG trends are discussed. These are compared to those from *Wöppelmann and Marcos* (2016). After that, the GNSS and ALT-TG trends are compared and optimal settings are discussed. For the comparison we take into account that both trends are not computed from time series covering the same period by correcting for non-linear VLM trends estimated from fingerprints.

### 3.1  Direct GNSS trends

For 570 TGs at least one GNSS station is found within a 50 km radius with an uncertainty on the trend that is below 1 mm $\text{yr}^{-1}$. The VLM for these TGs is shown in Fig. 2 using the median of the surrounding GNSS stations in case there are multiple trends available. The signature of GIA dominates the signal on large scales, and is primarily visible in Scandinavia and Canada.

**Table 2.** Statistics of trend differences between NGL and ULR5 at 70 stations for the eight approaches.

| | | RMS | Mean | Median |
|---|---|---|---|---|
| Approach | Keyword | mm yr$^{-1}$ | mm yr$^{-1}$ | mm yr$^{-1}$ |
| 1 | mean | 1.11 | 0.07 | 0.05 |
| 2 | median | 1.05 | 0.12 | 0.03 |
| 3 | closest | 1.36 | 0.02 | 0.02 |
| 4 | dist. weight. | 1.21 | 0.00 | 0.03 |
| 5 | longest | 1.29 | 0.32 | 0.20 |
| 6 | smallest error | 1.15 | 0.24 | 0.17 |
| 7 | error weight. | 1.11 | 0.08 | 0.02 |
| 8 | dist./error weight. | 1.23 | 0.01 | 0.05 |

**Table 3.** Number of TGs at which trends are estimated from differenced ALT-TG time series. The '-1.0' indicates no correlation threshold is set.

| Threshold | Number of TGs |
|---|---|
| -1.0 | 663 |
| 0.0 | 660 |
| 0.1 | 658 |
| 0.2 | 655 |
| 0.3 | 638 |
| 0.4 | 602 |
| 0.5 | 549 |
| 0.6 | 470 |
| 0.7 | 344 |

In Alaska there might be a significant contribution of present-day ice mass loss. If GIA is removed the VLM signals typically range between -3 and 3 mm yr$^{-1}$ (*Wöppelmann and Marcos*, 2016), with a few exceptions.

Even though the large-scale GIA process appears to be captured properly, regional VLM has a large effect on the GNSS trends. In Fig. 3 the differences between the lowest and highest VLM estimate from the eight methods discussed in Sect. 2.1.2 are shown. The extreme values primarily resulted from the 'mean', 'median' and 'inverse distance' methods (not shown). The figure shows that the range is generally higher, where more GNSS trends are available. In particular the seismically active zones like the US West Coast show a larger range. The range of solutions, when considering all TGs with at least two GNSS trends, has a mean of 0.92 mm yr$^{-1}$ with 25 and 75 percentiles of 0.38 and 1.20 mm yr$^{-1}$. In case at least three available GNSS trends are considered, the mean of the differences rises to 1.09 mm yr$^{-1}$ and the 25 and 75 percentiles to 0.56 and 1.34 mm yr$^{-1}$. Since we only considered GNSS trends with a maximum standard deviation of 1 mm yr$^{-1}$, this implies that a significant contribution of kilometer-scale VLM variations is present along the West Coast of the US, where the difference

between methods is often larger than 1 mm yr$^{-1}$. Note that the range of individual GNSS trends is on average even larger than the range between methods. *Santamaría-Gómez et al.* (2017) estimated the global numbers for the impact of spatial variations in VLM at 30 km and 100 km separation to be 0.2 mm yr$^{-1}$ and 0.5 mm yr$^{-1}$. At coasts of Europe and North America, where most tide gauges are located, these numbers are substantially larger, i.e. even the range between methods is on average larger than 1 mm yr$^{-1}$. The differences between methods is often comparable in size as the VLM signal, especially after the GIA is removed.

*Wöppelmann and Marcos* (2016) show that a comparison between their ALT-TG trends and their GNSS trends yields an RMS of 1.47 mm yr$^{-1}$. They use visual inspection to remove tide gauges where clear non-linear effects or discontinuities were present. In Table 2 a comparison is made between the eight different approaches and the GNSS trends of *Wöppelmann and Marcos* (2016) that were used in the aforementioned comparison with ALT-TG trends at 70 locations. The values show that a substantial fraction of the RMS between GNSS and ALT-TG trends can already be explained by different GNSS averaging and processing methods. Using the closest station (approach 3) an RMS of 1.36 mm yr$^{-1}$, which is comparable in magnitude to the RMS between GNSS and ALT-TG trends found by *Wöppelmann and Marcos* (2016). Note that we remove all NGL GNSS trends with an uncertainty larger than 1 mm yr$^{-1}$ and therefore co-located stations are sometimes removed. The closest GNSS station in our selection is therefore not always the same as the one used by *Wöppelmann and Marcos* (2016). The best comparison is found with the median (approach 2), even though the RMS of differences is still above 1 mm yr$^{-1}$. Since the closest station method depends on a single station, there is larger chance some outliers are present, which substantially increases the RMS of differences. For the closest station method three trend differences larger than 3 mm yr$^{-1}$ are found, whereas only one is found for the median method.

## 3.2 Differenced ALT-TG trends

Using correlation thresholds, we try to minimize the residual ocean signal in ALT-TG time series Additionally, it will filter problematic stations, where no correlation between TG and altimetry observations is found. A higher threshold reduces therefore the number of ALT-TG trends. Table 3 shows the reduction of the differenced VLM trends, when the correlation threshold increases. After a correlation threshold of 0.4, the number of observations drops substantially. At a threshold of 0.7, the number of TGs for which a trend is computed, is only half of that without a threshold. The remaining trends are generally more reliable, because of two reasons: VLM time series that exhibit relatively large residual ocean signals are removed; and secondly, TG time series that contain large jumps due to unidentified reasons (e.g. earthquakes or equipment changes) are removed.

In order to show that the method decreases the oceanic signal, we compare the standard deviation reduction by using correlation thresholds and weighting (Fig. 4). The plot in the top panel shows the comparison between the standard deviation of the differenced time series using no correlation threshold and the time series using a threshold of 0.7 together with a correlation weighting. The mean reduction in standard deviation is 3.9 mm, whereas the mean standard deviation is 37 mm. The change in standard deviations at several locations are coherent, which is expected because the sea level fluctuations along continental slopes are coherent (*Hughes and Meridith*, 2006). Substantial reductions in standard deviation are apparent at both North American coasts, in Japan and in Northern Europe. *Vinogradov and Ponte* (2011) had already observed large discrepancies in

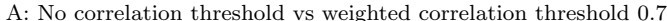

A: No correlation threshold vs weighted correlation threshold 0.7

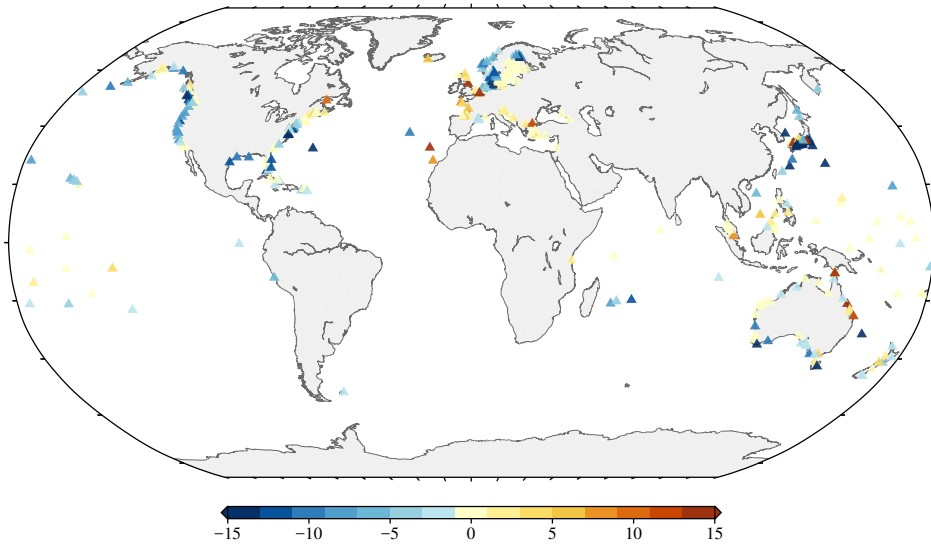

B: Unweighted correlation threshold 0.0 vs weighted correlation threshold 0.0

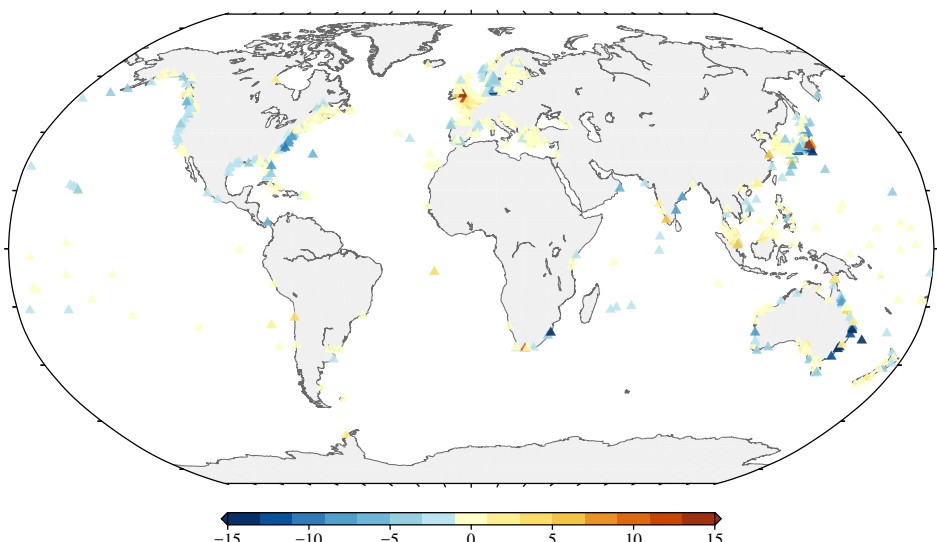

**Figure 4.** Change in standard deviation (mm) of the differenced time series using correlation thresholds and weighting. Note that a correlation threshold of 0.0 indicates positive correlations only.

interannual ocean signals between TGs and altimetry in North America and in Japan. It suggests that our technique is capable to reduce these ocean signals. This is confirmed by the change in the median of the spectral indices, $\kappa$, as discussed in Sect. 2.4. The median of the spectral indices changes from -0.63 to -0.57, which indicates that the autocorrelation in the residuals decreased. The Winter Harbour (Canada) VLM time series (Fig. 1) shows a typical example in which especially the correlated

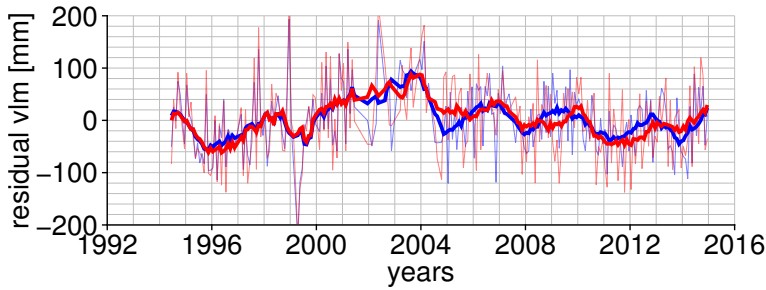

**Figure 5.** Time series of ALT-TG differenced VLM at the Llandudno (UK) TG. A moving-average filter is applied to visualize the interannual variability. In blue: with a threshold of 0.0 for the correlation, but without correlation weighting. In red: with a threshold of 0.0 for the correlation and with correlation weighting. In the background the time series without a moving-average filter applied.

noise is reduced. However, there are several locations where the standard deviation increases substantially. Most of them are sporadic, but in a few locations, like in the UK and France there is coherent increase.

Similar patterns of standard deviation decrease, albeit reduced in magnitude, are observed for the not-weighted against weighted VLM time series with a correlation threshold of 0.0 (bottom of Fig. 4), i.e. when only positively correlated altimetry

time series are taken into account. Instead of 344 VLM trends, as for the comparison discussed above, 660 trends are compared. The mean reduction of the standard deviation is 1.4 mm, whereas the mean standard deviation is 38 mm. Remarkable is the strong reduction of the standard deviation at the southeast side of Australia. In the UK and France an increase in standard deviation is present again. In most cases an increase in white noise, likely due to the decreased effective number of altimetry measurements, is responsible for the higher standard deviation, as demonstrated in Fig. 5 for a VLM time series at Llandudno,

UK. In most cases of an increasing standard deviation, the correlated ocean signals are still reduced or remain approximately equal.

Fig. 6 shows the VLM trends estimated from the ALT-TG time series using no correlation threshold and a threshold of 0.7. A comparison of Fig. 2 and Fig. 6 reveals that especially the Indian Ocean and the southern Pacific Ocean are sampled better using ALT-TG instead of GNSS trends. If the correlation threshold is set to 0.7, the number of trend estimates decreases,

which has particularly an impact on the number of trend estimates at TGs in South America and Africa. Hence, for regional reconstructions, a careful choice should be made for the correlation threshold.

Compared with the GNSS trends, the neighbouring ALTG-TG trends show more variation, which is especially true for the UK and Japan. It is difficult to say whether this is a true VLM signal, but it is important to note that many GNSS stations are placed on bedrock, which exhibits more stable trends than the coastal locations of tide gauges. Secondly, the GNSS trends

with an uncertainty larger than 1 mm yr$^{-1}$ are removed, which reduces the variability. Of the 663 ALT-TG trends, 293 (44 %) have a trend uncertainty smaller than 1 mm yr$^{-1}$. Therefore larger spatial trend variability can also be induced by remaining ocean signals in the VLM time series. In the Fig. 6B, showing the 0.7 threshold trends, the number of trends is reduced due to the correlation threshold. It removes most tide gauges in the highly variable regions mentioned before and the neighbouring

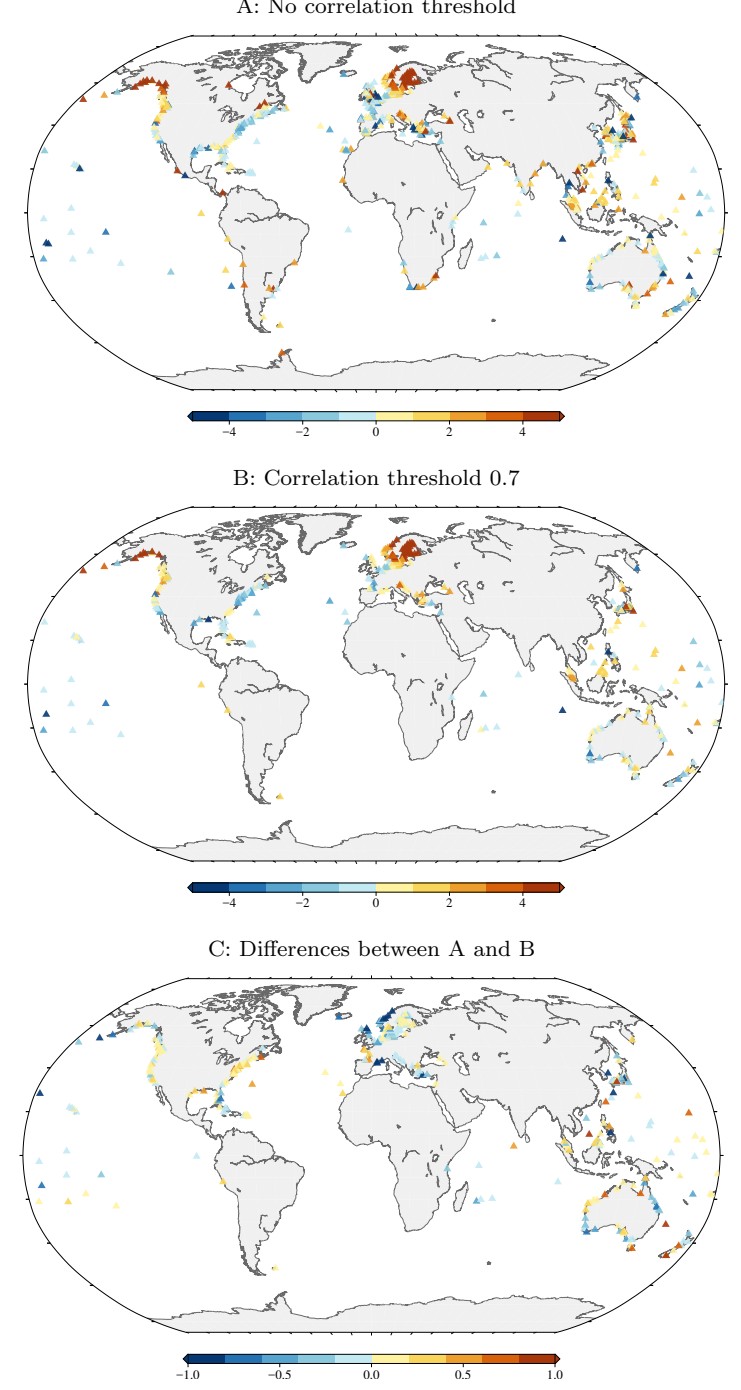

**Figure 6.** ALT-TG trends (mm yr$^{-1}$) estimated using no threshold (A), with a correlation threshold and correlation weighting (B) and the difference between them (C).

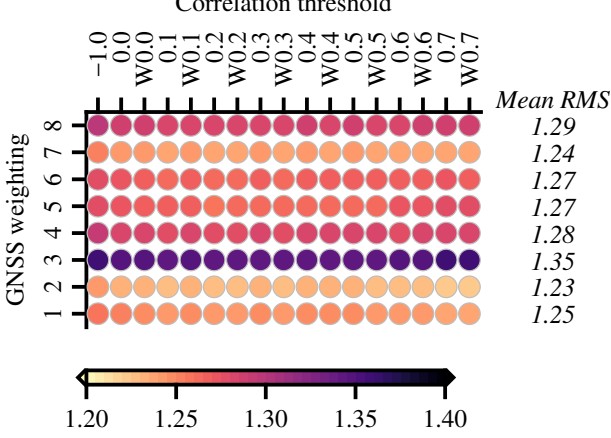

**Figure 7.** RMS (mm/yr) of differences between GNSS and ALT-TG VLM trends. The 'W' indicates weighting by correlation. The '-1.0' indicates no correlation threshold is set. The numbers of the y-axis refer to the approaches used to combine the GNSS trends as described in Sect. 2.1.2.

differences are therefore less erratic. 284 out of 344 trends (83 %) have a trend uncertainty smaller than 1 mm yr$^{-1}$ using the 0.7 correlation threshold.

The results of applying correlation weighting and thresholding are shown Fig. 6C. Two spots of coherent changes in the trends can be clearly identified: in Norway the trends increased by approximately 1 mm yr$^{-1}$, while in the East Coast of the

5  United States the opposite happens. These spots exhibit longshore coherent sea level signals that are not found in the open ocean (*Calafat et al.*, 2013; *Andres et al.*, 2013). Note that both locations also exhibit a strong reduction in standard devation (Fig. 4). Coherent changes are also present around Denmark. Other regions, where substantial reductions in the standard deviation are found, do not experience coherent changes in trends.

### 3.3  GNSS vs ALT-TG trends

10  In this section the VLM trends from GNSS using the eight approaches as described in Sect. 2.1.2 are compared with the differenced ALT-TG VLM trends using various correlation thresholds. Based on the intercomparison we determine the best solution for the GNSS approach and the correlation thresholds for altimetry. Additionally, a comparison is made with *Wöppelmann and Marcos* (2016). We also investigate the effect of present-day mass redistribution on the difference in trends due to varying time spans of the GNSS and the ALT-TG methods.

15  Fig. 7 shows the RMS of trends differences between various GNSS combination methods and correlation thresholds for ALT-TG. The RMS of trend differences is computed at 155 TG stations for which all solutions are available. The colors exhibit small differences horizontally and large differences vertically, indicating that the GNSS method is more important in reducing the RMS. The difference between the method with the lowest RMS of differences, which is obtained by taking the median of the GNSS trends (2), and the method with the highest RMS, which uses the closest GNSS station (3), is approximately 0.12

**Table 4.** Statistics of the differences between the median of the GNSS trends (approach 2) and the ALT-TG trends for various correlation thresholds. The 'W' indicates that the altimetry time series are weighted by the correlation. The row 'W&M' shows the comparison with *Wöppelmann and Marcos* (2016) trends. The column 'NoT' indicates the number TGs for which trend estimates are computed. On the left side of the table all stations are taken into account, on the right side only stations are taken into account for which a solution exist for all correlations thresholds (and including those from W&M).

| Correlation | All | | | | Same | | | |
| | RMS | Mean | Median | NoT | RMS | Mean | Median | NoT |
| | mm yr$^{-1}$ | mm yr$^{-1}$ | mm yr$^{-1}$ | | mm yr$^{-1}$ | mm yr$^{-1}$ | mm yr$^{-1}$ | |
|---|---|---|---|---|---|---|---|---|
| -1.0 | 2.141 | -0.241 | -0.107 | 294 | 1.234 | -0.167 | -0.099 | 137 |
| 0.0 | 2.108 | -0.248 | -0.101 | 294 | 1.226 | -0.175 | -0.068 | 137 |
| 0.0W | 2.103 | -0.250 | -0.036 | 294 | 1.219 | -0.172 | -0.056 | 137 |
| 0.1 | 2.113 | -0.258 | -0.096 | 293 | 1.219 | -0.174 | -0.074 | 137 |
| 0.1W | 2.108 | -0.260 | -0.043 | 292 | 1.218 | -0.170 | -0.045 | 137 |
| 0.2 | 2.082 | -0.233 | -0.073 | 292 | 1.217 | -0.163 | -0.074 | 137 |
| 0.2W | 2.080 | -0.234 | -0.015 | 292 | 1.216 | -0.168 | -0.042 | 137 |
| 0.3 | 1.986 | -0.152 | 0.047 | 283 | 1.221 | -0.157 | -0.066 | 137 |
| 0.3W | 1.991 | -0.157 | 0.056 | 283 | 1.217 | -0.165 | -0.044 | 137 |
| 0.4 | 1.695 | -0.106 | 0.065 | 264 | 1.223 | -0.152 | -0.050 | 137 |
| 0.4W | 1.696 | -0.112 | 0.071 | 264 | 1.218 | -0.158 | -0.041 | 137 |
| 0.5 | 1.554 | -0.086 | 0.044 | 239 | 1.220 | -0.153 | -0.058 | 137 |
| 0.5W | 1.552 | -0.087 | 0.056 | 239 | 1.217 | -0.155 | -0.067 | 137 |
| 0.6 | 1.417 | -0.093 | -0.065 | 204 | 1.209 | -0.155 | -0.087 | 137 |
| 0.6W | 1.416 | -0.093 | -0.083 | 204 | 1.208 | -0.156 | -0.094 | 137 |
| 0.7 | 1.220 | -0.142 | -0.123 | 155 | 1.206 | -0.140 | -0.060 | 137 |
| 0.7W | 1.220 | -0.144 | -0.124 | 155 | 1.206 | -0.142 | -0.074 | 137 |
| W&M | 1.658 | -0.177 | -0.050 | 211 | 1.328 | -0.101 | 0.020 | 137 |

mm yr$^{-1}$. *Hamlington et al.* (2016) computed VLM trends at TG locations by using a complex filtering procedure that also implicitly takes into account the median of the GNSS trends. Next to taking the median of the GNSS trends, taking the mean (1) within the 50 km radius and using variance weighting (7) also yield substantially lower RMS differences than the other five methods. However, the median method performs slightly better. Besides, the median method is less sensitive to large values caused by GNSS trends with larger uncertainties (for which the mean method is sensitive) and also less to outliers caused by large local VLM differences (for which the variance weighting method is sensitive).

In Table 4 we analyze the results for different correlation thresholds in more detail by comparing them to the GNSS trends based on the median method. On the left side of the table the RMS, mean and median are shown for all VLM estimates available for each correlation threshold. Setting no correlation thresholds yields trend estimates at 294 TGs for comparison,

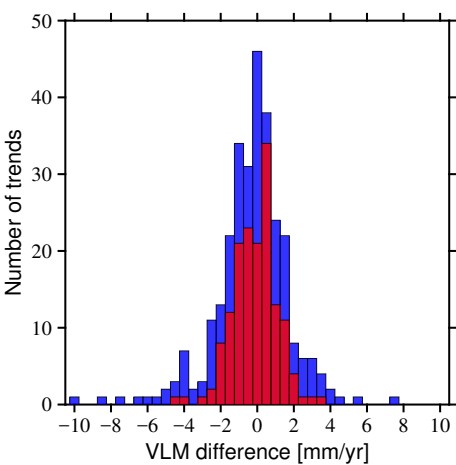

**Figure 8.** Histogram of GNSS and ALT-TG trend differences. In blue the results without any correlation threshold and in red with a correlation threshold of 0.7 and correlation weighting.

while setting a threshold at 0.7 leaves only 155. While the number of trends decreases, the RMS decreases as well, indicating that the correlation thresholds can serve as a selection procedure, which filters out outliers. This is confirmed by Fig. 8, in which we see the decrease of the number of available trends, but also the removal of the outliers. If the threshold is set to 0.7 only three discrepancies in trends of larger than 3 mm yr$^{-1}$ are found. Note that the reduction in RMS is not only caused by the removal of problematic ALT-TG time series. Large earthquakes for example might induces jumps or non-linear behaviour in both the TG and GNSS time series, so the larger range in Fig. 8 for no correlation threshold may be partly attributed to problematic GNSS trends. In the last row the *Wöppelmann and Marcos* (2016) trends are compared with our GNSS trends. It has a similar RMS with the 0.4-0.5 correlation threshold trends, but it is computed with a substantially smaller number of trends.

On the right side of the table, we only included TGs for which all solutions are available, which reduces the number from 155 to 137, because W&M trends are also considered for comparison. The RMS of differences for 155 stations is only slightly larger as will be shown below in Table 5. Note that the RMS of the residuals using ALT-TG from W&M, is already 0.14 mm yr$^{-1}$ lower than those in the study of *Wöppelmann and Marcos* (2016) and about 0.4 mm yr$^{-1}$ less than in *Pfeffer and Allemand* (2016), who incorporated only 109 and 113 stations, respectively. This is a consequence of the combined use of the median of the NGL trends and the selection based on correlation. Our altimetry solutions further decrease the RMS by another 0.1 mm yr$^{-1}$ compared to W&M, even when no threshold on the correlation is set. In the study of *Wöppelmann and Marcos* (2016), the along-track altimetry ALT-TG trends performed worse than the AVISO results. The reason for this discrepancy could be the latitudinal intermission bias, or the small radius around the TG used in that study for including altimetry measurements.

Increasing the correlation threshold only slightly reduces the RMS between GNSS and ALT-TG trends and the additional weighting has a neglectable effect on the RMS. As mentioned before, the threshold increase and correlation weighting generally reduced the standard deviation (Fig. 4) of the ALT-TG time series and Fig. 6 showed coherent changes in trend. Additionally,

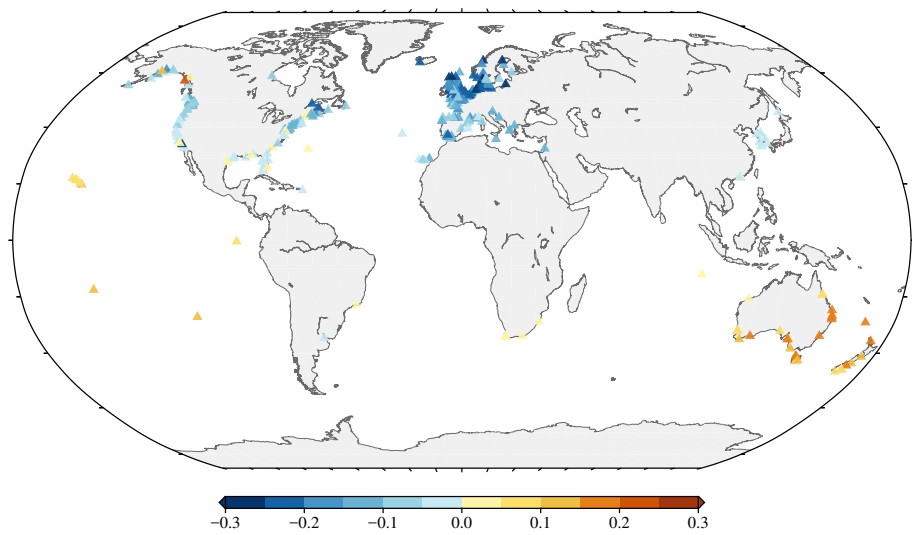

**Figure 9.** Trend differences (mm yr$^{-1}$) between the GNSS and ALT-TG time spans induced by non-linear VLM due to present-day mass redistribution.

the NGL and ULR trends showed an RMS of differences and range between the GNSS approaches of more than a millimeter. We argue that the absence of a clear improvement or a change in RMS due to correlation thresholds is a result of the relatively large noise in the GNSS trends. The histogram in Fig. 8 shows that for 155 stations, only three discrepancies are larger than 3 mm yr$^{-1}$. For these TGs (located at Galveston (US), Eureka (US) and the Cocos Islands (Australia)) we find that the neighbouring GNSS stations are located at the other side of lagoons or on different islands. Therefore the likely cause for the largest discrepancies is not the ALT-TG trend, but local VLM differences between the GNSS stations and the TG.

The third column of Table 4 shows that the mean is in all cases negative, i.e. the GNSS trends are larger than those of ALT-TG. Trends obtained with correlations -1.0, 0.0, 0.1 and 0.2 are barely statistically different from zero based on a 95% confidence level, while the others are not. The 95 % confidence level is taken as two times the standard deviation of the mean of the residual trends ($\frac{\sigma_n}{\sqrt{N}}$, where N is the number of trends and $\sigma_n$ the standard deviation of the residual trends). In the right 'mean' column for the 137 stations, the means are statistically insignificantly different from zero at the 95%-confidence level, wheras at a 90%-confidence level several are not. The medians in both columns are closer to zero and deviate up to 0.2 mm yr$^{-1}$ from the mean, which indicates a slightly skewed distribution.

There is a non-linear VLM signal due to present-day mass loss in both GNSS and ALT-TG trends and since they cover different time spans this causes small systematic differences between trends. Due to the inhomogeneous distribution of the TGs and the spatial signal of non-linear VLM, this affects not only the mean, but also the skewness of the distribution. In Fig. 9 the trend differences between the GNSS and ALT-TG methods are visualized for all 294 stations. Most of the negative differences in trends are observed in Europe and parts of North-America, while positive differences in trends are observed in Australia. In Europe there is an uplift due to present-day mass loss, which increases over the last few years. Since the GNSS

**Table 5.** Statistics of ALT-TG trend differences with the median GNSS approach for various correlation settings after applying a correction for non-linear VLM.

| Correlation | NoT: 155 | | | NoT: 137 | | |
|---|---|---|---|---|---|---|
| | RMS | Mean | Median | RMS | Mean | Median |
| | mm yr$^{-1}$ | mm yr$^{-1}$ | mm yr$^{-1}$ | mm yr$^{-1}$ | mm yr$^{-1}$ | mm yr$^{-1}$ |
| -1.0 | 1.231 | -0.102 | -0.039 | 1.223 | -0.100 | 0.030 |
| 0.0 | 1.225 | -0.109 | -0.027 | 1.215 | -0.108 | 0.031 |
| 0.0 | 1.223 | -0.106 | 0.016 | 1.209 | -0.105 | 0.048 |
| 0.1 | 1.220 | -0.107 | -0.014 | 1.208 | -0.107 | 0.034 |
| 0.1 | 1.222 | -0.104 | 0.003 | 1.208 | -0.104 | 0.072 |
| 0.2 | 1.220 | -0.099 | 0.016 | 1.207 | -0.096 | 0.027 |
| 0.2 | 1.221 | -0.101 | -0.001 | 1.206 | -0.101 | 0.059 |
| 0.3 | 1.223 | -0.091 | 0.011 | 1.211 | -0.090 | 0.018 |
| 0.3 | 1.221 | -0.098 | -0.001 | 1.207 | -0.098 | 0.036 |
| 0.4 | 1.226 | -0.087 | 0.011 | 1.214 | -0.085 | 0.021 |
| 0.4 | 1.223 | -0.092 | 0.008 | 1.209 | -0.091 | 0.037 |
| 0.5 | 1.225 | -0.088 | 0.020 | 1.212 | -0.086 | 0.042 |
| 0.5 | 1.222 | -0.090 | 0.027 | 1.208 | -0.088 | 0.045 |
| 0.6 | 1.222 | -0.087 | -0.007 | 1.202 | -0.088 | 0.018 |
| 0.6 | 1.222 | -0.087 | -0.006 | 1.201 | -0.089 | 0.028 |
| 0.7 | 1.220 | -0.071 | 0.021 | 1.202 | -0.073 | 0.037 |
| 0.7 | 1.219 | -0.074 | 0.012 | 1.201 | -0.075 | 0.036 |

time series are generally shorter, they measure a larger uplift signal. By subtracting the present-day VLM that GNSS observes from altimetry observations, we obtain negative signals in Europe.

We applied a correction for the effect of present-day mass loss to the trends for the 155 stations for which a trend is found with all methods in Table 5. Similarly, this is done for the 137 stations, so that the results are comparable with Table 4. There is no significant reduction in RMS. The maximal deviation of the median from zero is 0.06 mm yr$^{-1}$ for the 155 stations and maximally 0.07 mm yr$^{-1}$ for the 137 stations, which is a reduction with respect to the values listed in Table 4. The mean is also reduced to approximately -0.1 mm yr$^{-1}$, which is statistically equal to zero. This result is at the level of the noise in the determination of the ITRF origin (*Santamaría-Gómez et al.*, 2017) and it is smaller than the 0.4 mm yr$^{-1}$ to which global mean sea level trends from altimetry are gauranteed (*Mitchum*, 2000). Unless it is proven that the altimeters are more stable and the uncertainties in the ITRF origin are reduced, a mean of trend differences closer to zero cannot be expected.

## 4 Conclusions

We presented new ways to estimate VLM at TGs from GNSS and differenced ALT-TG time series. A comparison is made between eight different methods to obtain VLM at the TG from NGL GNSS trends. The range of the trends between the approaches is at the same level as the standard deviations of the GNSS trends, with a mean of 0.92 mm yr$^{-1}$ and a median of

0.71 mm yr$^{-1}$. A comparison with the estimates of ULR5 (*Wöppelmann and Marcos*, 2016) at 70 stations yielded an RMS of at least 1.05 mm yr$^{-1}$. A comparison with ALT-TG showed that using the median of all neighbouring GNSS provided the best results.

For the ALT-TG trends we used along-track data from the Jason series of altimeters. At every 6 km along-track data were stacked, to create time series. The time series were low-pass filtered with a moving-average filter of one year and correlated

with low-pass filtered TG time series. An average or weighted monthly time series for altimetry was created taking into account only the time series corresponding to correlations above a threshold. The TG time series were subtracted from the average of monthly low-pass filtered altimetry time series to create a ALT-TG time series. Using the Hector software between 344 and 663 trends were computed from the ALT-TG time series, depending on the correlation threshold set.

The standard deviation of the ALT-TG time series was reduced on average by approximately 10 % when a correlation

threshold of 0.7 was used. Spatially coherent differences in trends between various thresholds are observed at the east coast of the US and in Norway. We argue that residual interannual ocean variability in ALT-TG time series can locally induce VLM trend biases, especially when time series are short. For 155 stations globally distributed, increasing the correlation threshold does not significantly affect the RMS of differences between GNSS and ALT-TG trends. However, the correlation threshold also works as a selection procedure. When considering 294 VLM estimates from GNSS and ALT-TG at the same TGs for comparison,

with no threshold the RMS of differences was 2.14 mm yr$^{-1}$, whereas an RMS of 1.22 mm yr$^{-1}$ was reached using 155 stations and a threshold of 0.7. This is a substantial improvement with respect to the 1.47 mm yr$^{-1}$ RMS of *Wöppelmann and Marcos* (2016) at 109 TGs, the best result so far. Note that increasing the threshold considerably reduces the number of time series in the southern hemisphere and therefore other thresholds might be better depending on the purpose.

The comparison with tide gauges also reveals that the trends from ALT-TG are biased low (similar to *Wöppelmann and*

*Marcos* (2016)), even though this is barely significant. Using mass redistribution fingerprints, a correction is applied for trend differences caused by non-linear behaviour of present-day mass changes. The RMS of differences is barely affected, but the mean of differences is changed from about -0.2 to -0.1 mm yr$^{-1}$, which is now statistically insignificant.

The trends in this publication (median GNSS and ALT-TG for all correlations) are provided in the supplementary material. The ALT-TG trends are accompanied by errors bars computed using the Hector software. The provided uncertainties for the

GNSS use the MAD from the median of the trends within 50 km scaled by 1.4826 (*Wilcox*, 2005), similar to the MIDAS algorithm. If only a single GNSS station is present, the MIDAS uncertainty is provided. If two GNSS stations are present and both trends are statistically equal, it takes the square-root of the mean of the GNSS variances to avoid very small error bars. When no correlation threshold is used 663 ALT-TG and 570 GNSS trends are available at 939 different TGs. By setting the correlation threshold to 0.7, the number of TGs, for which a trend is estimated, decreases to 759. Depending on the application,

the value of the threshold can be varied to find an optimum between the reliability and the number of TG for which a trend is estimated. If both GNSS and ALT-TG trends are available, we recommend to use GNSS trends, because of correlated residual ocean signals between various ALT-TG time series. However, if a large discrepancy ($> 3$ mm yr$^{-1}$) is found between the GNSS and ALT-TG trends, we recommend to use the ALT-TG trend, because the culprit is likely local VLM differences between the

TG and the GNSS stations. The GNSS - ALT-TG histogram for no correlation threshold reveals large discrepancies between the two methods up to 10 mm yr$^{-1}$. While the problems with ALT-TG trends are mostly resolved by setting a higher threshold, the GNSS trends might still require some inspection before they are used in sea level studies. A faster practice is to use trend uncertainties, that carry information about the linearity of the trends, and when the MAD is used as described above, also information about local VLM variability. However, when only one GNSS station is present the information about local VLM

variations is absent.

## Appendix A: Intermission biases

The latitude-dependent intermission biases are computed from 1/8 degree latitudinally averaged sea surface height differences between TOPEXPOSEIDON and Jason-1 (TP-J1) and Jason-1 and Jason-2 (J1-J2). For the TP-J1 bias four separate polygons are estimated for ascending tracks and four for the descending tracks, while for J1-J2 a single polygon is estimated. Depending

on the geophysical corrections and the processing of the altimetry data, not all parameters are statistically different from zero based on variances of the residuals. However, to be consistent with the study of *Ablain et al.* (2015), we maintain the polygons as such.

*Acknowledgements.* This study is funded by the Netherlands Organisation for Scientific Research (NWO) through VIDI grant 864.12.012 (Multi-Scale Sea Level (MuSSeL)).

The MIDAS GNSS trends are obtained from the Nevada Geodetic Laboratory (NGL).

http://geodesy.unr.edu/

The altimetry data are obtained from the Radar Altimetry Database System (RADS).

http://rads.tudelft.nl/rads/rads.shtml

Permanent Service for Mean Sea Level (PSMSL), 2017, "Tide Gauge Data", Retrieved 1 November 2016.

http://www.psmsl.org/data/obtaining/

We would like to thank Marta Marcos and Guy Wöppelmann for sharing their trend estimates.

We thank Alvaro Santamaría-Gómez and an anomymous reviewer for their thorough reviews that helped to improve this article.

**Table 6.** Values for the parameters of the latitudinal intermission bias correction. These numbers are added to the sea surface height anomalies of the respective satellites. TP asc. and TP desc. indicates the function variables that should be added to the ascending and descending tracks of TOPEX/POSEIDON using Eq. (4), respectively. J2 indicates the function variables to be used for Jason-2.

| Parameter | TP asc. Lat(deg) | TP asc. Value | TP desc. Lat(deg) | TP desc. Value | Jason-2 Lat(deg) | Jason-2 Value |
|---|---|---|---|---|---|---|
| $c_0$(mm) | (-66.2,-1.5) | 80.3 | (-66.2,-1.5) | 77.3 | (-66.2,66.2) | 98.1 |
| $c_1$(mm deg$^{-1}$) | | $-2.3 \cdot 10^{-1}$ | | $-1.7 \cdot 10^{-1}$ | | $-9.3 \cdot 10^{-2}$ |
| $c_2$(mm deg$^{-2}$) | | $-1.1 \cdot 10^{-2}$ | | $1.2 \cdot 10^{-3}$ | | $3.8 \cdot 10^{-3}$ |
| $c_3$(mm deg$^{-3}$) | | $-3.0 \cdot 10^{-4}$ | | $2.9 \cdot 10^{-4}$ | | $8.4 \cdot 10^{-7}$ |
| $c_4$(mm deg$^{-4}$) | | $-2.4 \cdot 10^{-6}$ | | $3.8 \cdot 10^{-6}$ | | $-7.6 \cdot 10^{-7}$ |
| $c_0$(mm) | (-1.5,0.2) | 83.8 | (-1.5,1.3) | 79.9 | | |
| $c_1$(mm deg$^{-1}$) | | 1.3 | | 2.4 | | |
| $c_2$(mm deg$^{-2}$) | | -1.3 | | $5.2 \cdot 10^{-1}$ | | |
| $c_3$(mm deg$^{-3}$) | | $-5.3 \cdot 10^{-1}$ | | | | |
| $c_4$(mm deg$^{-4}$) | | | | | | |
| $c_0$(mm) | (0.2,4) | 84.9 | (1.3,4) | 73.3 | | |
| $c_1$(mm deg$^{-1}$) | | $-8.0 \cdot 10^{-1}$ | | 13.7 | | |
| $c_2$(mm deg$^{-2}$) | | $-8.6 \cdot 10^{-1}$ | | -5.1 | | |
| $c_3$(mm deg$^{-3}$) | | $1.5 \cdot 10^{-1}$ | | $4.9 \cdot 10^{-1}$ | | |
| $c_4$(mm deg$^{-4}$) | | | | | | |
| $c_0$(mm) | (4,66.2) | 72.9 | (4,66.2) | 75.8 | | |
| $c_1$(mm deg$^{-1}$) | | $8.1 \cdot 10^{-1}$ | | $7.9 \cdot 10^{-1}$ | | |
| $c_2$(mm deg$^{-2}$) | | $-2.8 \cdot 10^{-2}$ | | $-3.3 \cdot 10^{-2}$ | | |
| $c_3$(mm deg$^{-3}$) | | $3.4 \cdot 10^{-4}$ | | $6.4 \cdot 10^{-4}$ | | |
| $c_4$(mm deg$^{-4}$) | | $-1.1 \cdot 10^{-6}$ | | $3.9 \cdot 10^{-6}$ | | |

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
