# Peer review of "A comparison of methods to estimate vertical land motion trends from GNSS and altimetry at tide gauge stations"

_Ocean Science, 2017_

## Referee Comment (RC1) · Anonymous Referee #1 · 13 Dec 2017

General comments:

Accurately determining vertical land motion at tide gauges is an important scientific issue with crucial societal implications associated with future relative sea levels at the coast. The study by Kleinherenbrink et al builds upon the most recent estimates of vertical land motion from GNSS data analyses and the combination of satellite altimetry and tide gauge data. The authors perform a detailed and honest critical review of the estimates available from the literature, while they provide ways to overcome some of the limitations. For instance, wherever there is no permanent GNSS antenna at the very top of the tide gauge (co-location), but multiple GNSS receivers are in the vicinity,

they explore different methods to deal with this situation. In addition, they delve into the details of the best possible way of deriving estimates from the combination of satellite altimetry and tide gauge data with insightful outcomes too.

The manuscript reflects a sound scientific approach. The methods applied are clearly outlined. Some minor technical details are missing, however, and require clarification (see below). The results are discussed in detail, and overall the results and discussion provide a substantial contribution to the area of research on determining vertical land motion at tide gauges. In addition, the manuscript is well structured, clear and concise, and the conclusions are supported by the data. A somewhat negative note is that I miss that the authors are not providing their best estimates on vertical land motion (with the error bars) in a supplemental material. Similar to the studies they build upon, they should provide their estimates for future investigation. Perhaps this can be considered by the authors for the final version. In conclusion, my suggestion is a minor revision before publication.

Other (minor or technical) comments:

p.1, Title: The term "weighting" does not correspond to several of the approaches examined in this study. See also 1st and 10th lines in the abstract). In addition, I would change "derive" to "estimate" to underline that behind the scenes the results from these methods are based on an estimation procedure, not directly observed.

p.1, Lines 2-3: It should be clarified that these methods are considered to deal with the situation of multiple GNSS stations nearby a TG.

p.1, Line 20: Ostanciaux et al. did not established the magnitude that can reach the GIA effect. I suggest to quote an original early reference such as Gutenberg, in Bull. geol. Soc. Am. (1941).

p. 1, Lines 21-22: The statement that trends at TGs are affected by erosion is not obvious to me. Please, quote a reference that demonstrates this relationship. p.3,

Line 26: For the sake of consistency, I wonder why Hector is not applied for the GNSS trends too. Can you develop the §with your arguments, please?

p.3, Line 31: The issue is primarily that the differential land motion between the GNSS antenna and the tide gauge is not monitored locally, for instance via repeated levelling campaigns. Thus, a lack of information.

p4. Line 7: I guess "However" is not correct here. Considering revisiting this since the decrease in accuracy is not associated with the use of the software and its advantages.

p4. Line 11: the term measurements is not appropriate here, the positioning time series are outcomes (estimates) of the measurements analysis.

p.4, Line 12: Please, develop how the scaling is performed (what is its origin).

p.4, Line 12: typo in "devations", should be "deviations"

Section 2.1.1: did you screen the GNSS time series for apparent transient processes that would impact (question the validity of) the linear trend estimation?

Section 2.1.2: See above my comment on the term "weighting". Within this section you use the term "approach" which is definitely more appropriate.

p.4, Line 17: at some point (here or later in the manuscript) you should discuss this vague statement "a record long enough".

p.5, Lines 3-5: You detailed the "obvious" relationship of method [7], you should detail that of method [8], which is less obvious to me.

p.5, Line 7: Holgate is published in 2013 (not 2012). See also reference list (p.22, Line 33).

p.5, Lines 22-25: Please, rephrase. I had to read the sentence several times. Consider splitting it into two sentences.

p.5, Line 30: Please, develop the rationale for 250 km (why not 200 km, or 270 km,
**OSD**

or. . .).

p.7, Table 2: The information conveyed by this table is too technical. Consider moving it to an Appendix or Supplemental material. Clarify what are these differences (related to J1? TP-J1, then J2-J1?). In addition, add error bars to the parameter estimates, and/or say if all these parameters are statistically significant at the 95% level.

p.8, Line 4: "are computed" should be "is computed".

p.9, Lines 6-7: The sentence has a problem. I don't understand, please rephrase.

p.9, Line 15: What is the rationale for the 50km radius. Please, develop.

p.10, Table 3: Consider adding a mnemonic keyword (after the number) to designate the approach, for instance "closest", "longest", etc.

p.13, Lines 7-8: Can you quantify the amount of reduction using equation (4)?

p.20, Line 2: Strictly speaking, "observations" is not appropriate (estimates? data?)

---

## Referee Comment (RC2) · A. Santamaría-Gómez (Referee) · 23 Jan 2018

**General comments:**

This paper addresses the methods of estimating the linear trend of the vertical land motion (VLM) at tide gauge (TG) stations using GPS and satellite altimetry minus TG observations (ALT-TG). Since the satellite altimetry and most of the GPS data are not provided at the TG location itself, the paper focuses on the different choices to extrapolate these datasets to the TG location. Some of the investigated choices have been used in past sea-level studies and are relevant for comparison purposes, whereas others (especially the treatment of satellite altimetry observations) are new in this paper. The comparison of the different choices provides valuable information to other scientists working on this subject and the preferred choices of the authors lead to a reduction of the VLM differences between GPS and ALT-TG techniques compared to previous studies.

The writing is clear globally, but some sentences (details given below) need clarification or correction. Parts of the methodology need also clarification (for instance, concerning the pole tide or the use of errorbars). The authors focus on describing the results of their analysis without going in depth with their discussion and implications, which undermines the conclusions to some extent.

Specific comments:

The title: I would suggest changing "data weighting methods" by "methods" or "approaches".

The abstract needs to be improved to make it more clear and self-contained. As it is now, it looks like a compressed listing of the results so it may be hard for the readers to understand without a minimal background and way out (recommendations or take-home messages).

P1L18: several VLM processes are modelled. This is very ambiguous. It may be true (we could model anything), but only GIA models are actually being used. Later on, it's said that local VLM processes cannot be captured by models.

P2L11: Actually, Santamaría-Gómez et al. 2017 did not conclude on the accuracy, but they show bigger differences between ULR and NGL than ULR and the other solutions being compared. Compared to the other solutions, NGL velocities also had larger errorbars to accommodate these differences.

P3L16-20: In addition to the ocean signal, the ALT-TG correlation can be used to infer the correlation between the TG record and the VLM \*of\* the TG itself, especially with low-pass filtered series as you did, for instance if the VLM at the TG is not linear

OSD
during the altimetry period. This is inseparable from the ocean signal you mention (see discussion in Santamaría-Gómez et al. 2014, JoGE).

P4L5-7: Note that all the ULR solutions have been computed using the CATREF software (Altamimi et al. 2016). CATS has been used to re-estimate the trend uncertainty, but the estimated trend does not change statistically. The "slight change in trend" comes together with the increased uncertainty and it is just the consequence of inverting a more complex covariance matrix in time with probably a small contribution also from the different use of spatial covariance between CATREF and CATS.

P4L29: I assume the approaches using the longest time series (5) and the smallest error (6) are also using the closest GPS station, but it would be better to clarify.

P5L3: It would be better to add here the equation of the approach (8) to see how the approaches (4 distances) and (7 weighted mean) are combined. To me, this is in theory the best approach since it uses more information available that the other approaches. However, the way the distance and uncertainty are combined may still be very important. Also, the propagation of the VLM uncertainty from the GPS to the TG should be commented on as it varies for each approach.

Section 2.3: I am confused about which altimetry series did you use and when. You say that an "additional" filtered set was used to test interannual correlation (P6L10-12) and that before estimating the correlation, you removed residual seasonal cycles (P6L15). So, where do these residual seasonal cycles come from if the series were filtered? Is the yearly moving-average not enough to remove unmodeled Sa tides from altimetry or the low-pass filter allows for annual variations at the TG? Finally, the filtered series were kept for the analysis (P7L1), but Figures 1 and 5 show both filtered and unfiltered series.

P6L13-15: Is it necessary to remove the ocean pole tide from the ALT and TG records? Are they significantly different? Concerning the solid Earth pole tide, I would suggest adding that the RADS solid Earth pole tide model is consistent with the Desai's model

OSD
concerning a linear mean pole trajectory, so that the interannual vertical deformation is preserved in the TGs when subtracting one and adding the other (I assume this was the purpose, but it could be said explicitly). However, what is the rationale for adding the IERS solid Earth pole tide to the TG records after removing the RADS model (P8L1-3)? Contrary to the RADS or Desai's models, the IERS solid Earth pole tide model does not correct the interannual deformation (see King and Watson, 2014). The interannual deformation was removed by the RADS model and is not restored by the IERS model. In doing so, the ALT-TG VLM will not be consistent with the GPS VLM that is still affected by this interannual deformation from the IERS model. If I understood your treatment, I think you should add the Desai's model in both cases.

P8L5-8: note that the IERS conventions were updated about this issue in June 2015, and even though the issue still persists, most of the GPS VLM estimates are based on the old IERS implementation, at least the ULR and NGL solutions you used. The 0.1 mm/yr error arises in a regional sea-level reconstruction using GPS-corrected TG records with old IERS model. The VLM effect at individual GPS sites may be 3 times larger (King and Watson, 2014). Explain how this error is corrected using the mass redistribution fingerprints.

Section 2.4 could be integrated into the 2.3

P9L10: change ULR by ULR5, which is the solution used by Wöppelmann and Marcos, 2016

Section 3.1 and elsewhere: direct/indirect are ambiguous terms. I would suggest using GNSS and ALT-TG for consistency.

Figure 3 and elsewhere: change spread by range

P10L7: change solutions by weighting methods for consistency or even to approaches, which may be more appropriate.

P11L1-4: The range values are driven by the extremes, which are obtained from the
"mean", "median" and "inverse distance" approaches. None of these approaches is using the information provided by the VLM errorbars, which can be as large as 1 mm/yr, and only the "median" approach is less affected by outlier VLM values (but only if we have a large sample and we assume the VLM estimates in 50 km follow a Gaussian distribution, which may not). I would suggest using the interquantile range instead of the range to evaluate the dispersion of the different approaches.

P11L6-7: In relation to my comment before, these global estimates of spatial variations of VLM were given as 1 sigma standard deviations. You would have to multiply them by 5 or more to obtain something close to the range of the extremes (for instance, by 10 in areas with strong GIA gradient). On top of that, a global figure will never fit all locations which will be underestimated or overestimated.

P11L11-16: Table 3 shows the VLM differences at 70 TGs between using the closest ULR5 value and 8 different approaches with the NGL velocities. It is surprising that the RMS of the differences is the highest for the closest NGL value (approach 3), which will use the same GPS station as in ULR5 for many TGs, whereas it is minimum for the median of the NGL values 50 km around the TG (approach 2). The WRMS of the differences between ULR6 and NGL is about 0.7 mm/yr. You are using ULR5 and not ULR6 here, but the RMS for the closest NGL station is two times larger and appears unreasonable to me. It may be due to the VLM errorbars not being used. Also the ranking of the methods in this table and that in Figure 7 matches exactly as if the ULR5 velocities were providing the same benchmarking information as the ALT-TG trends. Is this coincidental?

Figure 4: Change "reduction" by "change" or invert the sign of the scale for consistency (positive reduction is good, otherwise is bad).

Figure 7: It would be easier to read the legend if the mean RMS of each line, with fairly constant values, is added on the right of the figure, for instance.

P16L6-8: Please explain how the median takes into account the standard deviation of
the GNSS trends as in a weighted mean (approach 7). Also, any approach using more than one GNSS trend in 50 km around the TG is filtering the spatial variations in VLM, including the variance weighting (weighted mean) approach. From these lines on, it is decided that the median approach is the best candidate, but I'm not fully convinced and I would suggest adding more discussion on these results. For instance, the fact that a simple median provides better results than the more complex approach of including distance and uncertainty information needs better discussion. The combination of the distance and errorbar information is not trivial and may depend on the TG location, so this may have flawed this approach. However, even the weighted mean is using additional relevant information, but it is ranked after the median and the mean. This makes me think whether the evaluation using the ALT-TG trends is the best benchmark. For instance, the ALT-TG VLM uncertainties are probably large as well, with important variations among the TGs (correlation, length of the series, etc), and it seems to me that they were not used for the benchmarking either. On the other hand, the alternative explanation would be that the trend uncertainties of the NGL solution are not providing a useful value of their precision. For instance, it is known that there are trend biases not explained by their formal uncertainty and caused by a combination of the time series length and non-linear effects like seasonal signals, discontinuities, interannual deformation, transients, etc. Different processes would also bias the ALT-TG trends (orbital error, altimeter bias drift, etc.).

P2L20 and elsewhere: the correct reference for the ULR5 solution is Santamaría-Gómez et al 2012 Glob. Planet Change.

I fully agree with the last sentence and I would add that, whenever possible, one should always inspect the data being used. A much extended (and faster) practice is always using the trend uncertainties together with the trends, because they (should) carry relevant information on the linearity of the observed series.

---

## Author Comment (AC1) · 1 Feb 2018

**Rebuttal – reviewer 1**

*We would like to thank the anonymous reviewer for the thorough review of our manuscript. We update the manuscript such that it answers the questions and implements the recommendations of the reviewer. Below we wrote a point-by-point response to the reviewer comments.*

**General comments:**
**Accurately determining vertical land motion at tide gauges is an important scientific issue with crucial societal implications associated with future relative sea levels at the coast. The study by Kleinherenbrink et al builds upon the most recent estimates of vertical land motion from GNSS data analyses and the combination of satellite altimetry and tide gauge data. The authors perform a detailed and honest critical review of the estimates available from the literature, while they provide ways to overcome some of the limitations. For instance, wherever there is no permanent GNSS antenna at the very top of the tide gauge (co-location), but multiple GNSS receivers are in the vicinity, they explore different methods to deal with this situation. In addition, they delve into the details of the best possible way of deriving estimates from the combination of satellite altimetry and tide gauge data with insightful outcomes too.**

**The manuscript reflects a sound scientific approach. The methods applied are clearly outlined. Some minor technical details are missing, however, and require clarification (see below). The results are discussed in detail, and overall the results and discussion provide a substantial contribution to the area of research on determining vertical land motion at tide gauges. In addition, the manuscript is well structured, clear and concise, and the conclusions are supported by the data. A somewhat negative note is that I miss that the authors are not providing their best estimates on vertical land motion (with the error bars) in a supplemental material. Similar to the studies they build upon, they should provide their estimates for future investigation. Perhaps this can be considered by the authors for the final version. In conclusion, my suggestion is a minor revision before publication.**
*We were already planning to make the data publicly available. The vertical land motion estimates for all altimetry-tide gauge correlation settings and the median GNSS approach are now provided in the supplementary material. The aforementioned sentence is adjusted accordingly.*
*The technical details and other comments are discussed below.*

**Other (minor or technical) comments:**

**p.1, Title: The term "weighting" does not correspond to several of the approaches examined in this study. See also 1st and 10th lines in the abstract). In addition, I would change "derive" to "estimate" to underline that behind the scenes the results from these methods are based on an estimation procedure, not directly observed.**
*We changed the title to: "A comparison of methods to estimate vertical land motion trends from GNSS and altimetry at tide gauge stations." The text is adjusted as well, so 'approach' is used.*

**p.1, Lines 2-3: It should be clarified that these methods are considered to deal with the situation of multiple GNSS stations nearby a TG.**

*Two sentences are added at the beginning of the abstract and the existing text is adjusted accordingly. "Global Navigation Satellite System (GNSS) are usually not co-located with Tide Gauges (TGs). Therefore trends from neighbouring GNSS stations are combined to estimate a VLM trend at the TG. This study compares eight methods to estimate Vertical Land Motion (VLM) trends at 570 TG stations using GNSS."*

**p.1, Line 20: Ostanciaux et al. did not established the magnitude that can reach the GIA effect. I suggest to quote an original early reference such as Gutenberg, in Bull. geol. Soc. Am. (1941).**
*The reference to Ostanciaux is replaced with the one to Gutenberg.*

**p. 1, Lines 21-22: The statement that trends at TGs are affected by erosion is not obvious to me. Please, quote a reference that demonstrates this relationship.**
*We changed the sentence. Erosion and gas extraction are removed. So now the sentence reads: "including water storage, postseismic deformation and anthropogenic activities (references)."*

**p. 3, Line 26: For the sake of consistency, I wonder why Hector is not applied for the GNSS trends too. Can you develop the §with your arguments, please?**
*We wanted to have the best possible GNSS trends for which we do not have to apply any screening. The trends are strongly affected by jumps in the time series. Based on the Blewitt et al. (2016) their MIDAS method has the smallest equivalent step size detection. Therefore we selected this method. We cannot apply the same method to the altimetry-tide gauge time series, since the time series cover only ~200 months, which is rather short for the MIDAS approach. We rephrased several sentences in Sect. 2.1.1 to clarify the reasoning.*

**p.3, Line 31: The issue is primarily that the differential land motion between the GNSS antenna and the tide gauge is not monitored locally, for instance via repeated levelling campaigns. Thus, a lack of information.**
*A sentence is added that addresses this issue.*

**p4. Line 7: I guess "However" is not correct here. Considering revisiting this since the decrease in accuracy is not associated with the use of the software and its advantages.**
*We rephrased the sentence and split it up in two separate sentences. "The software is also able to estimate and detect discontinuities that occur due to earthquakes and equipment changes. Even though a large fraction of the trend estimates have formal accuracies better than 1 mm/yr, undetected discontinuities might significantly bias the estimated trends (Gazeaux et al., 2013)."*

**p4. Line 11: the term measurements is not appropriate here, the positioning time series are outcomes (estimates) of the measurements analysis.**
*The term measurements is replaced by estimates.*

**p.4, Line 12: Please, develop how the scaling is performed (what is its origin).**
*We added an equation and a reference to Wilcox (2005).*

**p.4, Line 12: typo in "devations", should be "deviations"**
*Updated.*

**Section 2.1.1: did you screen the GNSS time series for apparent transient processes that would impact (question the validity of) the linear trend estimation?**
*No, we did not screen the time series, because we use the pre-computed trends from MIDAS. Any non-linear behavior might bias the trend, but it will also inflate the error bars as described in the section. We now clearly state that we do not apply any screening.*

**Section 2.1.2: See above my comment on the term "weighting". Within this section you use the term "approach" which is definitely more appropriate.**
*The term weighting is either replaced by 'method' or by 'approach' throughout the manuscript.*

**p.4, Line 17: at some point (here or later in the manuscript) you should discuss this vague statement "a record long enough".**
*We changed the sentence, such that: "currently only a few have a record that ensures a trend accuracy of 1 mm/yr or less".*

**p.5, Lines 3-5: You detailed the "obvious" relationship of method [7], you should detail that of method [8], which is less obvious to me.**
*An equation is added for method [8].*

**p.5, Line 7: Holgate is published in 2013 (not 2012). See also reference list (p.22, Line 33).**
*The reference is updated.*

**p.5, Lines 22-25: Please, rephrase. I had to read the sentence several times. Consider splitting it into two sentences.**
*The sentence is rephrased.*

**p.5, Line 30: Please, develop the rationale for 250 km (why not 200 km, or 270 km, or ...).**
*Outside of the equatorial regions and the continental shelves, ocean correlation scales are below 250 km (Ducet et al. 2000; Roemmich et al. 2009), so we do not expect significant improvements if observations outside of the 250 km range are included. We could probably find some long-shore correlation along the shelves over longer distances, but it would not be appropriate to take those observations into account, since long-term trends do not have to resemble anymore, i.e. large-scale signals like GIA trends are not equal to the TG location anymore. On top of that, at least one track of the altimeters is always passing through the 250 km region. Making the radius smaller, reduces the number of observations substantially, especially at lower latitudes.*

**p.7, Table 2: The information conveyed by this table is too technical. Consider moving it to an Appendix or Supplemental material. Clarify what are these differences (related to J1? TP-J1, then J2-J1?). In addition, add error bars to the parameter estimates, and/or say if all these parameters are statistically significant at the 95% level.**
*We moved the table to the appendix.*
*The caption is extended, to clarify what the differences mean.*
*Since we do not apply a full error propagation on these values, it is difficult to determine whether they are statistically significant or not. If we use the variances (sigma\*2) of the residuals to compute the, then the errors for the coefficient $c=sigma^2*(A^T*A)^{-1}$, several of coefficients are not statistically significant (primarily in the equatorial regions for TP-J1). This is primarily, because we average*

*the altimetry differences per latitude band (1/8 degree wide) first and then compute the polynomials. The degrees of freedom (~10 dof) is therefore rather small to estimate proper statistics for the equatorial regions. We therefore stick to the polynomials as used in the Ablain's paper, which is referenced to in the text.*

**p.8, Line 4: "are computed" should be "is computed".**
*Updated.*

**p.9, Lines 6-7: The sentence has a problem. I don't understand, please rephrase.**
*The sentence is rephrased.*

**p.9, Line 15: What is the rationale for the 50km radius. Please, develop.**
*Most studies involving sea level include observations within radii of 10-100 km. We took the radius right in the middle, but we could have increased or descreased the radius. A radius of 100 km would include observations with errors due to local VLM of more than 0.5 mm/yr on average (Santamaria-Gomez et al., 2017), while taken a small range reduces the number of trends substantially. Tests, however, demonstrated that similar results are obtained for 30 and 70 km, but with slightly less or more trends estimates, respectively. Several sentences are added in the methodolgy section.*

**p.10, Table 3: Consider adding a mnemonic keyword (after the number) to designate the approach, for instance "closest", "longest", etc.**
*We added keywords in the table.*

**p.13, Lines 7-8: Can you quantify the amount of reduction using equation (4)?**
*Yes, we can. The median of the spectral indices (for the same stations) is closer to zero for higher correlation settings. We added a line with the statistics.*

**p.20, Line 2: Strictly speaking, "observations" is not appropriate (estimates? Data?)**
*We guess this should be line 7. The term observations is replaced with data.*

---

## Author Comment (AC2) · 1 Feb 2018

**Rebuttal – reviewer 1**

*We would like to thank reviewer Alvaro Santamaria-Gomez for his comments that helped to improve and clarify the paper. We addressed the comments below and changed the manuscript accordingly.*

**General comments:**
**This paper addresses the methods of estimating the linear trend of the vertical land motion (VLM) at tide gauge (TG) stations using GPS and satellite altimetry minus TG observations (ALT-TG). Since the satellite altimetry and most of the GPS data are not provided at the TG location itself, the paper focuses on the different choices to extrapolate these datasets to the TG location. Some of the investigated choices have been used in past sea-level studies and are relevant for comparison purposes, whereas others (especially the treatment of satellite altimetry observations) are new in this paper. The comparison of the different choices provides valuable information to other scientists working on this subject and the preferred choices of the authors lead to a reduction of the VLM differences between GPS and ALT-TG techniques compared to previous studies.**

**The writing is clear globally, but some sentences (details given below) need clarification or correction. Parts of the methodology need also clarification (for instance, concerning the pole tide or the use of errorbars). The authors focus on describing the results of their analysis without going in depth with their discussion and implications, which undermines the conclusions to some extent.**

**Specific comments:**

**The title: I would suggest changing "data weighting methods" by "methods" or "approaches".**
*The title has been changed to: "A comparison of methods to estimate vertical land motion trends from GNSS and altimetry at tide gauge stations."*

**The abstract needs to be improved to make it more clear and self-contained. As it is now, it looks like a compressed listing of the results so it may be hard for the readers to understand without a minimal background and way out (recommendations or take-home messages).**
*We extended the abstract with several sentences where we focus on the recommendations that are given in the conclusions.*

**P1L18: several VLM processes are modelled. This is very ambiguous. It may be true (we could model anything), but only GIA models are actually being used. Later on, it's said that local VLM processes cannot be captured by models.**
*We removed the sentence. We added: "The large scale VLM processes, such as Glacial Isostatic Adjustment (GIA) and the elastic response of the Earth due to present-day mass redistribution can be modelled to accuracies close to 1 mm/yr. However, TG are often only corrected for the GIA signal, which typically ... ".*

**P2L11: Actually, Santamaría-Gómez et al. 2017 did not conclude on the accuracy, but they show bigger differences between ULR and NGL than ULR and the other solutions being compared. Compared to the other solutions, NGL velocities also had larger errorbars to accommodate these differences.**

*We changed the sentence to: "They concluded that the number of stations in the NGL database was larger, but that the differences between neighbouring stations was significantly larger than the Jet Propulsion Laboratory (JPL) and ULR6 solutions."*

**P3L16-20: In addition to the ocean signal, the ALT-TG correlation can be used to infer the correlation between the TG record and the VLM \*of\* the TG itself, especially with low-pass filtered series as you did, for instance if the VLM at the TG is not linear during the altimetry period. This is inseparable from the ocean signal you mention (see discussion in Santamaría-Gómez et al. 2014, JoGE).**

*While the largest interannual signals come from the ocean, time series affected by for example discontinuities caused by earthquakes or other non-linear VLM behavior are also removed, because they will not have a high correlation. We tried to address this point on page 17, but we added a part to the sentence on page 3 to make sure this is stated.*

**P4L5-7: Note that all the ULR solutions have been computed using the CATREF software (Altamimi et al. 2016). CATS has been used to re-estimate the trend uncertainty, but the estimated trend does not change statistically. The "slight change in trend" comes together with the increased uncertainty and it is just the consequence of inverting a more complex covariance matrix in time with probably a small contribution also from the different use of spatial covariance between CATREF and CATS.**

*We agree that "The slight change in trend" is not significant and it leads to confusion. Therefore this part of the sentence is removed.*

**P4L29: I assume the approaches using the longest time series (5) and the smallest error (6) are also using the closest GPS station, but it would be better to clarify.**

*No, they do not. They take the longest time series, or the time series with the smallest error within the 50 km radius. We changed the sentences to clarify this.*

**P5L3: It would be better to add here the equation of the approach (8) to see how the approaches (4 distances) and (7 weighted mean) are combined. To me, this is in theory the best approach since it uses more information available that the other approaches. However, the way the distance and uncertainty are combined may still be very important. Also, the propagation of the VLM uncertainty from the GPS to the TG should be commented on as it varies for each approach.**

*We added the equation for approach 8 and we agree that in theory approach 8 would be the best.  A possible reason that it does not perform best is probably the limited number of tide gauge for which we have multiple GNSS stations. The statistics are therefore obscured by several 'exceptional cases'. We can find multiple examples in the Netherlands where a close distance is not a good indicator for the representativeness of the VLM at the tide gauge, mainly due to anthropogenic effects, for example local gas extraction or groundwater fluctuations.*
*Uncertainties are not provided directly, because it would in several variants not respresent a proper estimation of the real uncertainty. For example, the weighting used in method 5 (longest) would just give you the GNSS trend uncertainty, but it does not include any relative local VLM information in the uncertainty. It is therefore definitely underestimated. We also do not use any uncertainty information in the rest of the manuscript. In the additional material we provide the trends and uncertainties and distance to the GNSS stations from the tide gauges. On top of that we provide the best (median) solutions with error bars. We added a line about this in the*

*conclusions. If somebody wishes to compute the uncertainties for the other methods, it can easily be done by propagating the uncertainties of the individual GNSS trends.*

**Section 2.3: I am confused about which altimetry series did you use and when. You say that an "additional" filtered set was used to test interannual correlation (P6L10-12) and that before estimating the correlation, you removed residual seasonal cycles (P6L15). So, where do these residual seasonal cycles come from if the series were filtered? Is the yearly moving-average not enough to remove unmodeled Sa tides from altimetry or the low-pass filter allows for annual variations at the TG? Finally, the filtered series were kept for the analysis (P7L1), but Figures 1 and 5 show both filtered and unfiltered series.**

*The time series are filtered with a moving-average low-pass filter of a year. A moving-average low-pass filter does not completely remove semi-(annual) cycles, which you can see by the response in the frequency domain. An alternative would be to use a different filter, but this leaves larger transient zones at the beginning and the end of the time series. We added a sentence to clarify this.*

*It is slightly confusing indeed. Basically we create two low-pass filtered time series: one monthly, one yearly. The yearly one is used to determine interannual correlation. If the yearly ones have a correlation with the TG time series higher than the correlation threshold, the corresponding monthly ones are kept. The other ones are removed. In the figure we averaged the remaining monthly ones, and then low-pass filtered them to show that the interannual signals are reduced. We have rewritten several sentences and updated the caption to clarify this.*

**P6L13-15: Is it necessary to remove the ocean pole tide from the ALT and TG records? Are they significantly different? Concerning the solid Earth pole tide, I would suggest adding that the RADS solid Earth pole tide model is consistent with the Desai's model concerning a linear mean pole trajectory, so that the interannual vertical deformation is preserved in the TGs when subtracting one and adding the other (I assume this was the purpose, but it could be said explicitly). However, what is the rationale for adding the IERS solid Earth pole tide to the TG records after removing the RADS model (P8L1-3)? Contrary to the RADS or Desai's models, the IERS solid Earth pole tide model does not correct the interannual deformation (see King and Watson, 2014). The interannual deformation was removed by the RADS model and is not restored by the IERS model. In doing so, the ALT-TG VLM will not be consistent with the GPS VLM that is still affected by this interannual deformation from the IERS model. If I understood your treatment, I think you should add the Desai's model in both cases.**

*The ocean pole tide can be as large as 2 cm and will therefore affect the correlation parameters. The RADS pole tide includes contributions from the solid Earth, loading and ocean tide tides. The altimeter is affected by all three. The tide gauge is only affected by the ocean tide. We subtract however the full RADS pole tide from both of the time series. To be consistent, we have to add the loading and the solid Earth tide. The loading tide is very small (typically 10 % of the ocean tide), so it barely has an effect on the correlation and therefore we ignore this term (especially after filtering). The solid Earth tide amounts to almost a centimeter and even though it will probably not affect the correlation, we add it back to be safe. In both cases Desai's model is used. The interannual deformation, or the non-linear part of the mean pole, is therefore corrected for in both cases.*

**P8L5-8: note that the IERS conventions were updated about this issue in June 2015, and even though the issue still persists, most of the GPS VLM estimates are based on the old IERS implementation, at least the ULR and**

**NGL solutions you used. The 0.1 mm/yr error arises in a regional sea-level reconstruction using GPS-corrected TG records with old IERS model. The VLM effect at individual GPS sites may be 3 times larger (King and Watson, 2014). Explain how this error is corrected using the mass redistribution fingerprints. Section 2.4 could be integrated into the 2.3.**

*We therefore apply the old solution to get the ALT-TG trends in the same system as the NGL solutions. This means that the trends can differ as mentioned in King and Watson, 2014) due to the non-linear drift of the pole. The non-linear drift of the pole is primarily caused by the melting of the ice sheets at locations away from the rotation axis of the Earth (mostly Greenland), which is captured by the sea level equation (it includes rotation changes). Several sentences are added to section 2.3 to clarify this.*

*We keep section 2.4 and 2.3 separate, because in the results we discuss the solutions without the present-day mass redistribution correction first separately.*

**P9L10: change ULR by ULR5, which is the solution used by Wöppelmann and Marcos, 2016**

*Updated.*

**Section 3.1 and elsewhere: direct/indirect are ambiguous terms. I would suggest using GNSS and ALT-TG for consistency.**

*We changed the terms to GNSS and ALT-TG.*

**Figure 3 and elsewhere: change spread by range**

*The term range is used instead of spread.*

**P10L7: change solutions by weighting methods for consistency or even to approaches, which may be more appropriate.**

*The term weighting method is not used anymore in combination with GNSS trends throughout the manuscript.*

**P11L1-4: The range values are driven by the extremes, which are obtained from the "mean", "median" and "inverse distance" approaches. None of these approaches is using the information provided by the VLM errorbars, which can be as large as 1 mm/yr, and only the "median" approach is less affected by outlier VLM values (but only if we have a large sample and we assume the VLM estimates in 50 km follow a Gaussian distribution, which may not). I would suggest using the interquantile range instead of the range to evaluate the dispersion of the different approaches.**

*Instead of providing the mean and the median, we now give the mean and the 25-75 % percentiles.*

**P11L6-7: In relation to my comment before, these global estimates of spatial variations of VLM were given as 1 sigma standard deviations. You would have to multiply them by 5 or more to obtain something close to the range of the extremes (for instance, by 10 in areas with strong GIA gradient). On top of that, a global figure will never fit all locations which will be underestimated or overestimated.**

*We adjusted the sentence and removed the word underestimation.*

**P11L11-16: Table 3 shows the VLM differences at 70 TGs between using the closest ULR5 value and 8 different approaches with the NGL velocities. It is surprising that the RMS of the differences is the highest for the closest NGL value (approach 3), which will use the same GPS station as in ULR5 for many TGs, whereas it is minimum for the median of the NGL values 50 km**

**around the TG (approach 2). The WRMS of the differences between ULR6 and NGL is about 0.7 mm/yr. You are using ULR5 and not ULR6 here, but the RMS for the closest NGL station is two times larger and appears unreasonable to me. It may be due to the VLM errorbars not being used. Also the ranking of the methods in this table and that in Figure 7 matches exactly as if the ULR5 velocities were providing the same benchmarking information as the ALT-TG trends. Is this coincidental?**

*We remove all GNSS trends with uncertainties larger than 1 mm/yr. This removes GNSS stations that might be co-located with the tide gauge. If then the closest station is used from NGL, it might be that it is not the same station as used by ULR5. In that case the closest station method depends on a single station not co-located with the tide gauge and therefore it is likely that some outliers are present. From the 70 stations, we find three trend differences larger than 3 mm/yr for the closest station method, while only one for the median method. We added a short discussion on this matter.*

**Figure 4: Change "reduction" by "change" or invert the sign of the scale for consistency (positive reduction is good, otherwise is bad).**
*We changed the word reduction to change.*

**Figure 7: It would be easier to read the legend if the mean RMS of each line, with fairly constant values, is added on the right of the figure, for instance.**
*We added the mean RMS in the figure.*

**P16L6-8:**
**[A] Please explain how the median takes into account the standard deviation of the GNSS trends as in a weighted mean (approach 7). Also, any approach using more than one GNSS trend in 50 km around the TG is filtering the spatial variations in VLM, including the variance weighting (weighted mean) approach.**
**[B] From these lines on, it is decided that the median approach is the best candidate, but I'm not fully convinced and I would suggest adding more discussion on these results. For instance, the fact that a simple median provides better results than the more complex approach of including distance and uncertainty information needs better discussion. The combination of the distance and errorbar information is not trivial and may depend on the TG location, so this may have flawed this approach.**
**[C] However, even the weighted mean is using additional relevant information, but it is ranked after the median and the mean. This makes me think whether the evaluation using the ALT-TG trends is the best benchmark. For instance, the ALT-TG VLM uncertainties are probably large as well, with important variations among the TGs (correlation, length of the series, etc), and it seems to me that they were not used for the benchmarking either.**
**[D] On the other hand, the alternative explanation would be that the trend uncertainties of the NGL solution are not providing a useful value of their precision. For instance, it is known that there are trend biases not explained by their formal uncertainty and caused by a combination of the time series length and non-linear effects like seasonal signals, discontinuities, interannual deformation, transients, etc. Different processes would also bias the ALT-TG trends (orbital error, altimeter bias drift, etc.).**

*[A] Suppose that we have no relative vlm movement in the area. Then the expected value between the tide gauge vlm trend and the observed GNSS trends is zero. The*

*GNSS trends with larger uncertainties are likely to have a deviation further from zero. Therefore it is likely that the median value is closer to the GNSS trends with smaller uncertainties. Besides, any 'outlying' values do not affect the median. Now suppose that the have relative vlm movement in the area. When variance weighting is applied, and the GNSS station with the large relative difference has by coincidence the lowest variance, it will get the highest weight, while actually it is the worst proxy. In the case of median weighting, the median will not be affected by this outlier. We rephrased the sentence, such that: "The median method is less sensitive to large values caused by GNSS trends with larger uncertainties (for which the mean method is sensitive) and also less to outliers caused by large local VLM differences (for which the variance weighting method is sensitive)."*

*[B] The distance-variance weighting approach does is more sensitive to the distance than to the variance, especially because the maximum uncertainty for the GNSS trends is set to a maximum of 1 mm/yr. For example, a trend found at 10 kilometer distance is already 10 times weaker than one at 1 km, while the uncertainties have often similar values, mostly 0.7-1 mm/yr. Therefore it effectly reduces the number of GNSS trends being used. We added a note in the methods section that the method is strongly depending on the distance. As a recommendation, we mention that using another distance weighting method might be better, but that it would require information of VLM correlation distances to find an optimum.*

*[C] The VLM uncertainties where indeed not used for benchmarking, because the uncertainties for the GNSS methods are not trivial and probably do not properly respresent the true uncertainty, because the uncertainty information due to relative VLM between GNSS and the TG is not present. As mentioned in [A], the weighted mean can be strongly affected by outliers due to local VLM differences. Just to give an indication of the uncertainties of the ALT-TG time series, we added some statistics to the ALT-TG results. Of the 663 trends for no correlation threshold, 293 have an uncertainty smaller than 1 mm/yr. Of the 344 trends for a correlation threshold of 0.7, 284 trends have an uncertainty smaller than 1 mm/yr.*

*[D] The MIDAS method takes annual differences in vertical location, so the seasonal signals are reduced to the minimum. Interannual deformation, transients, etc. will widen the distribution of the annual differences. Therefore the uncertainties increase. Since we use a maximum of 1 mm/yr on the uncertainty, it will therefore remove all trends computed from time series with substantial earthquake activity (see for example Japan) or interannual signals, like groundwater storage. We added a line in the GNSS methods section to state this.*

*Altimeter stability is guaranteed up to 0.4 mm/yr. If the altimeter would really be drifting with 0.4 mm/yr, this would increase the mean of the ALT-TG vs GNSS differences, but this has only a 0.06 mm/yr effect on the RMS. Temporally varying orbital errors would show up in the ALT-TG time series, so their contribution is captured in the uncertainty estimates produced by Hector.*

**P2L20 and elsewhere: the correct reference for the ULR5 solution is Santamaría-Gómez et al 2012 Glob. Planet Change.**
*The reference is added.*

**I fully agree with the last sentence and I would add that, whenever possible, one should always inspect the data being used. A much extended (and faster) practice is always using the trend uncertainties together with the trends, because they (should) carry relevant information on the linearity of the observed series.**
*We added a final sentence that the error bars carry relevant information about the non-linearity of the time series, but when a single station is present it does not contain any information about local VLM variability.*